# Aggregate Models, Not Explanations: Improving Feature Importance Estimation

**Joseph Paillard** [1] [2]    **Angel Reyero Lobo** [2] [3]    **Denis A. Engemann** [1]    **Bertrand Thirion** [2]

## Abstract

Feature-importance methods show promise in transforming machine learning models from predictive engines into tools for scientific discovery. However, due to data sampling and algorithmic stochasticity, expressive models can be unstable, leading to inaccurate variable importance estimates and undermining their utility in critical biomedical applications. Although ensembling offers a solution, deciding whether to explain a single ensemble model or aggregate individual model explanations is difficult due to the nonlinearity of importance measures and remains largely understudied. Our theoretical analysis, developed under assumptions accommodating complex state-of-the-art ML models, reveals that this choice is primarily driven by the model's excess risk. In contrast to prior literature, we show that ensembling at the model level provides more accurate variable-importance estimates, particularly for expressive models, by reducing this leading error term. We validate these findings on classical benchmarks and a large-scale proteomic study from the UK Biobank.

## 1. Introduction

Machine-learning (ML) models have become more complex to address the challenge of high-dimensional data across heterogeneous biomedical datasets, including multi-omics, medical imaging, and large-scale electronic health records. This has pushed the boundaries of biomedical research. At the same time, explainability methods have emerged, promising to translate this predictive power into biological and medical insights (Mi et al., 2021; Novakovsky et al.,

[1]Roche Pharma Research & Early Development, Roche Innovation Center Basel, F. Hoffmann-La Roche Ltd, Basel, Switzerland [2]Universite Paris-Saclay, Inria, CEA, Palaiseau, France [3]Institut de Mathematiques de Toulouse, UMR5219 Universite de Toulouse, France. Correspondence to: Joseph Paillard <joseph.paillard@roche.com>.

*Proceedings of the 43rd International Conference on Machine Learning*, Seoul, South Korea. PMLR 306, 2026. Copyright 2026 by the author(s).

2023). Such methods, sometimes also referred to as explainable AI (XAI), include feature importance measures that quantify the contribution of input variables and can help identify the drivers of the underlying data-generating mechanism (Shmueli, 2010; Ewald et al., 2024); hereafter, we use the terms feature importance and explanation interchangeably. Feature importance methods have high potential for scientific discovery; for example, in a model that predicts disease risk from multi-omic data, they can enable the identification of therapeutic targets and the development of biomarkers for prognosis or treatment response. However, the overparameterization and stochastic training inherent to complex models introduce significant *instability*, making them sensitive to hyperparameters, optimization stochasticity, and data sampling (Bottou & Bousquet, 2007). Although there are strategies to maintain predictive performance despite such instability, the consequences for feature importance estimation are not well understood. Notably, when this instability yields feature importance estimates that vary across models, it diminishes their usefulness in critical biomedical applications. For instance, this occurs when making decisions about investing in the experimental validation of therapeutic targets or implementing public health measures for identified risk factors. Therefore, simple models are often preferred to characterize underlying biological mechanisms due to their intrinsic interpretability. Directly analyzing coefficients in a linear model or rules in a decision tree, for instance, is typically favored over applying explainability methods to expressive architectures (Watanabe et al., 2023).

This instability, in which models with similar performance provide conflicting explanations of the same data, is known as the *Rashomon effect* (Donnelly et al., 2023). It was named after the classic Japanese film Rashomon, which depicts conflicting witness accounts of the same incident during a trial (Kurosawa, 1950). Addressing this instability is essential in high-stakes settings where inconsistent explanations can undermine the reliability and adoption of expressive architectures. This problem has consequently received substantial attention, with research focusing on aggregating information across multiple models to mitigate dependence on any single instance (Fisher et al., 2019; Donnelly et al., 2023). While this line of work demonstrates that aggregating importance scores derived from individual

models improves the stability of explanations, there remains a lack of formal analysis characterizing the feature importance estimation error. Our theoretical analysis identifies the primary error terms in this process and establishes the effectiveness of the alternative strategy: estimating importance from a model-level ensemble that aggregates predictions, rather than the standard approach of aggregating individual explanations. Importantly, due to the nonlinear nature of feature importance measures, these two approaches are not equivalent. This work aims to systematically study these aggregation strategies to improve the robustness of model-agnostic feature importance estimation. Our contributions are twofold:

- **Theoretical analysis**: We analyze feature importance estimation error under relaxed assumptions suitable for modern ML models. This framework enables a rigorous comparison of aggregation strategies leading to actionable insights and provides an error decomposition that formalizes the concepts of *stability* and the *Rashomon effect*.

- **Empirical results**: We perform a systematic comparison of ensembling approaches across established benchmarks using expressive architectures. We demonstrate the framework's utility on high-dimensional data from a large-scale proteomic cohort in the UK Biobank.

Our analysis covers Leave-One-Covariate-Out (*LOCO*, subsection 4.1), marginal and conditional Shapley Additive Global importancE (*SAGE*, subsection 4.2), Conditional Feature Importance (*CFI*, subsection 4.3), Permutation Feature Importance (*PFI*, subsection 4.4), and Integrated Gradients (*IG*, Appendix E), revealing that the benefits of aggregating models vary across feature importance measures.

## 2. Problem Setting

**Notations**  Let $(X, Y)$ be paired random variables following an unknown distribution $P$. We consider a supervised learning model $f_{\theta, \mathcal{D}_n}$ trained to predict $Y$ from $X$, where the subscripts $\mathcal{D}_n$ and $\theta$ represent the model's dependence on the training set of size $n$ and algorithmic randomization (e.g., initialization), respectively. We define the expected risk of a model $f$, for a loss $\mathcal{L}$, as $\mathcal{R}(f) = \mathbb{E}[\mathcal{L}(f(X), Y)]$, and its empirical risk on a test set as $\mathcal{R}_n(f)$. For simplicity, we assume that the test set is of comparable size $O(n)$, and we denote $n$ the overall sample complexity of the experiment. The excess risk of a model $f$ is defined as $\mathcal{E}(f) = \mathcal{R}(f) - \mathcal{R}(f_\star)$, where $f_\star$ is the risk minimizer over the data distribution. We omit the dependence of $\mathcal{E}$ on the model when the context is unambiguous. Finally, $\psi(f, j)$ denotes the importance of the $j^{th}$ feature for model

$f$. For conciseness, we denote the true importance (derived from $f_\star$) as $\psi_\star(j)$ and the empirical estimate from a trained model as $\psi_n(j)$.

Many variable importance measures (VIMs) rely on risk differences. For instance, *LOCO* compares the risk of a model, $f_\star^{-j}$ restricted to the subset of features excluding the $j^{th}$, to that of the full model. That is

$$\psi_\star^{\mathrm{loco}}(j) = \mathcal{R}(f_\star^{-j}) - \mathcal{R}(f_\star). \tag{1}$$

It estimates the total Sobol index, a well-studied metric for quantifying the predictive contribution of a variable (Homma & Saltelli, 1996). CFI targets the same quantity. Other measures of importance, such as *SAGE*, rather compare risks of restricted models over all possible feature subsets. Consequently, analyzing the error in estimating the risk of a given model is a prerequisite for understanding the behavior of these importance measures in real data. To this end, Williamson et al. 2023 decompose the risk estimation error as follows:

$$\mathcal{R}_n\left(f_{\theta, \mathcal{D}_n}\right) - \mathcal{R}(f_\star) = \tag{2}$$
$$\underbrace{[\mathcal{R}_n(f_\star) - \mathcal{R}(f_\star)]}_{A} + \underbrace{[\mathcal{R}(f_{\theta, \mathcal{D}_n}) - \mathcal{R}(f_\star)]}_{\mathcal{E}} + r_n$$

where $r_n = [\mathcal{R}_n(f_{\theta, \mathcal{D}_n}) - \mathcal{R}(f_{\theta, \mathcal{D}_n})] - [\mathcal{R}_n(f_\star) - \mathcal{R}(f_\star)]$, represents a second-order remainder term, which is shown to be negligible under standard assumptions. The first term, $A$, captures the dependence of the error on the finite test set, as it is an approximation of the expected risk in the absence of error in the model. The second term, the excess risk $\mathcal{E}$, quantifies the discrepancy between the trained model and the oracle risk minimizer, capturing errors stemming from the training process.

Following the framework of Bottou & Bousquet 2007, the excess risk can be decomposed into three components:

$$\mathcal{E} = \mathcal{E}_{\mathrm{app}} + \mathcal{E}_{\mathrm{est}} + \mathcal{E}_{\mathrm{opt}} \tag{3}$$

which are illustrated in Figure 1. First, the approximation error, $\mathcal{E}_{\mathrm{app}}$, captures how well the chosen function class $\mathcal{F}$ can approximate the data-generating process $f_\star$. Second, the estimation error, $\mathcal{E}_{\mathrm{est}}$, measures the impact of learning on a finite training set $\mathcal{D}_n$. This term accounts for sampling *stability*, reflecting that changes in the training data can result in a different estimation error. It can be mitigated using *bagging*, which has been shown to improve convergence rates for random forests (Biau & Scornet, 2016) and reduce variance through a smoothing effect (Bühlmann & Yu, 2002). Lastly, the optimization error, $\mathcal{E}_{\mathrm{opt}}$, reflects the impact of algorithmic stochasticity (parameterized by $\theta$) on the excess risk. This term is closely linked to the *Rashomon effect* because arbitrary optimization choices, such as random weight initialization or feature splits, lead to distinct

equally predictive models, and consequently distinct importances. It can be addressed by aggregating predictions from models trained with different randomizations (Allen-Zhu & Li, 2020), which we refer to as *voting*.

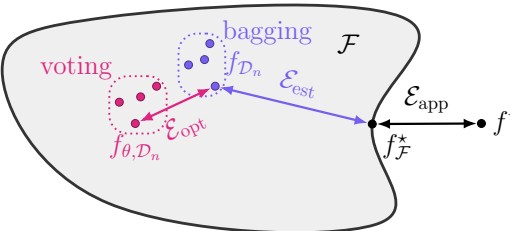

*Figure 1.* **Approximation, estimation, optimization trade-off.** To estimate the data-generating process $f_\star$ using a function class $\mathcal{F}$, we first account for the approximation error $\mathcal{E}_{\text{app}}$ inherent to the best possible function $f_{\mathcal{F}}^*$ in the class. Minimizing empirical risk over a finite training set $\mathcal{D}_n$ introduces estimation error $\mathcal{E}_{\text{est}}$, resulting in $f_{\mathcal{D}_n}$, which can be reduced via bagging. Finally, the stochasticity of the learning procedure adds optimization error $\mathcal{E}_{\text{opt}}$, leading to the final estimate $f_{\theta,\mathcal{D}_n}$. This error is mitigated by ensembling over multiple random initializations (voting).

A frequent critique of variable importance analysis is its dependence on a specific model instance, which hinders inference about the underlying data-generating process. Consequently, different models—arising from sampling instability or algorithmic stochasticity—may yield conflicting explanations despite achieving similar predictive performance. To move beyond the limitations of a single model, one might naturally turn to ensembling; however, this raises a practical question: **Should we derive feature importance from a model-level ensemble, or aggregate the importance scores of individual sub-models?** Due to the non-linearity of common loss functions, such as mean squared error (MSE) or cross-entropy, used to define importance metrics, these two strategies are not equivalent in general. Formally, let $f_{\text{ens}} = \frac{1}{B} \sum_{b=1}^{B} f_b$ denote an ensemble constructed from individual models $\{f_b\}_{b=1}^{B}$, where the subscript $b$ captures the dependence of each constituent model on its specific training sample $\mathcal{D}_{n,b}$ and algorithmic randomization $\theta_b$. For classification tasks, the output of each model $f_b$ consists of predicted class probabilities. In general, the *sub-models* strategy, $\frac{1}{B} \sum_{b=1}^{B} \psi(f_b, j)$, which aggregates the importances of individual models, is not equivalent to the *ensemble* strategy, $\psi(f_{\text{ens}}, j)$, which measures the importance of the model-level ensemble. This distinction reveals a fundamental trade-off in estimation accuracy. While the *sub-models* approach averages individual statistics to reduce the variance of the estimate, it cannot reduce the individual excess risks that contribute to the bias. In contrast, the *ensemble* approach acts directly on these error terms by providing a more accurate predictor, thereby reducing the bias.

## 3. Related Work

Feature importance methods aim to quantify the contribution of variables within an underlying data-generating process. While examining the coefficients of linear models offers an intuitive approach, their limited expressivity often fails to capture complex relationships. Consequently, we focus on *model-agnostic* feature importance methods. Furthermore, many popular techniques, such as Mean Decrease in Impurity (MDI) or permutation importance, are heuristics. These methods have been shown to be inconsistent as they do not converge asymptotically to a meaningful population-level quantity (Scornet, 2020). We therefore restrict our analysis to methods that estimate well-defined theoretical indices (Reyero-Lobo et al., 2025). Specifically, we consider Leave-One-Covariate-Out (*LOCO*, Williamson et al. 2023) and Conditional Feature Importance (*CFI*, Chamma et al. 2023). These metrics estimate the total Sobol index and facilitate the identification of the *Markov blanket*, which is the minimal set of variables required for optimal prediction (Bénard et al., 2022). We also examine Shapley Additive Global Importance (*SAGE*, Covert et al. 2020), a game-theoretic approach that identifies all variables linked to the outcome.

In their framework for *model-agnostic* importance, Williamson et al. 2023 primarily focus on test-set variance (term $A$ in Equation 2), by assuming the excess risk is asymptotically negligible, i.e. $\mathcal{E} = o_P(n^{-1/2})$. Specifically, to justify this point, the proofs of Theorems 1 and 2 in Williamson et al. 2023 require a *minimum rate of convergence* (Assumption B1) and Theorem 1 additionally requires that the model belongs to a Donsker class, hence further limiting its complexity. Precisely, denoting $|| \cdot ||_2$ the $L_2(P)$ over the data distribution $P$, the *minimum rate of convergence* requires that $||f_{\theta,\mathcal{D}_n} - f_\star||_2 = o_p(n^{-1/4})$. However, as detailed in Bach 2024, the convergence rates of overparametrized neural networks typically suffer from the curse of dimensionality, scaling at rates slower than $n^{-1/(d+2)}$ for dimension $d$. Similarly, Biau & Scornet 2016 indicates that random forests exhibit comparable behavior regarding dimensionality. While the existence of a lower-dimensional latent space may mitigate these effects, it remains unlikely that the convergence rates for these models satisfy the $o_P(n^{-1/4})$ requirement. By relying on these strict convergence hypotheses, the practical reach of the *model-agnostic* framework is technically narrowed, as its guarantees may not extend to highly expressive algorithms. In section 4, we present an alternative analysis that relies on weaker assumptions that are more realistic for modern machine learning (ML) algorithms. This analysis shows that the estimation error is, in fact, governed by $\mathcal{E}$, the error from the ML model.

The various sources of model estimation error presented in Equation 3 can result in discrepancies in feature im-

portance estimates. In the worst case, this may lead to inconsistent importance rankings or unreliable feature selection. To mitigate this, Donnelly et al. 2023 proposed the Rashomon Importance Distribution (RID), a framework that derives importance distributions across the "Rashomon set" (Fisher et al., 2019) of well-performing models. Setting aside the difficulty of estimating distributions over entire model classes, which has only been done for simple models, this strategy (referred to as *sub-models*) aggregates the importance scores of individual models. However, it has not been compared to the *ensemble* alternative, which consists of ensembling predictions at the model level and evaluating the importance of the resulting, more accurate ensemble. Furthermore, the lack of consensus on a formal definition of *stability* in this literature (Donnelly et al., 2023) makes it difficult to establish a rigorous comparison between these two strategies, which may explain why model-level ensembling has been overlooked. While *stability* often echoes the idea of variance, the ultimate goal is to accurately estimate feature importance; therefore, bias should not be ignored. We prefer to use the total estimation error, which captures both components.

Prior work has demonstrated that ensembling can be particularly effective at reducing model error, thereby addressing bias in variable importance estimates. Allen-Zhu & Li 2020, for instance, demonstrated that for overparametrized neural networks addressing a multi-view problem, employing an ensemble approach substantially reduces the optimization error component. A multi-view problem is characterized by data where the support consists of multiple features; however, for a small fraction of samples, a specific "view" (feature of the support) does not contribute to the data-generating process (see Definition 3.3 in Allen-Zhu & Li 2020). Specifically, in the context of a $k$-class classification problem on such data, Allen-Zhu & Li (2020) demonstrate that an ensemble of a small number of neural networks (typically fewer than 10) can dramatically reduce the optimization error. We hypothesize that this sharp reduction in individual model error through the *ensemble* strategy leads to a significant reduction in bias compared to the *sub-models* approach, therefore improving feature importance estimation.

## 4. Theoretical analysis

In this section, we identify the components contributing to the estimation error of VIMs that rely on risk differences. This analysis enables a rigorous comparison between two distinct strategies: measuring the importance of a single ensemble predictor (*ensemble*) or aggregating the importance values derived from individual predictors (*sub-models*). This comparison highlights the critical error terms to target for improving importance estimation efficiency,

especially when the excess risk of the model dominates the total error due to slow convergence rates. Our theoretical findings are subsequently validated through empirical evaluations across diverse datasets. The detailed proofs of all results can be found in Appendix A.

### 4.1. *LOCO*

As defined in Equation 1, *LOCO* consists in a difference of risks. We therefore start by deriving a result on the impact of finite sample error on the estimation of the Bayes risk $\mathcal{R}(f_\star)$. To relax the constraints on model complexity imposed by prior work (see section 3) and to present a theory that encompasses modern ML algorithms, we rely on two weaker assumptions.

**Assumption 4.1** (*Loss consistency*). $f_{\theta,\mathcal{D}_n}$ is a minimizer of the risk: $||\mathcal{L}(f_{\theta,\mathcal{D}_n}(X), Y) - \mathcal{L}(f_\star(X), Y)||_2 = o_P(1)$

**Assumption 4.2** (*Finite variance of the loss*). The loss evaluated at the risk minimizer $f_\star$, has a finite variance with respect to the data distribution: $\mathrm{Var}\left(\mathcal{L}(f_\star(X), Y)\right) < \infty$.

Assumption 4.1 reflects the intuitive idea that to provide a relevant measure of importance, a model must first be effective at the prediction task and thereby capture the underlying data-generating process. Assumption 4.2 is a mild condition, also adopted by Bottou & Bousquet 2007. Notably, it is weaker than the sub-Gaussian assumption standard in statistical learning, which requires the existence of all moments. For instance, for additive noise, $Y = f_\star(X) + \epsilon$ with the MSE loss, this assumption only imposes that $\mathrm{Var}(\epsilon^2) < \infty$.

**Proposition 4.3** (Bayes risk estimation error). *Under Assumptions 4.1 and 4.2, we have that*

$$\mathcal{R}_n\left(f_{\theta,\mathcal{D}_n}\right) - \mathcal{R}(f_\star) = \mathcal{E} + O_P(n^{-1/2}). \qquad (4)$$

*Proof sketch.* The proof follows the error decomposition presented in Equation 2. We first bound the remainder term $r_n$, by applying Lemma 1 from Kennedy (2024) which leads to $r_n = O_P\left(||\mathcal{L}(f_{\theta,\mathcal{D}_n}(X), Y) - \mathcal{L}(f_*(X), Y)||_2/\sqrt{n}\right)$. Assumption 4.1 *(loss consistency)* then ensures that this remainder is $r_n = o_P(n^{-1/2})$. Finally, we observe that the term A comes from the need to estimate the expectation of the loss using a finite test set. Note that Assumption 4.2 *(finite variance of the loss)* enables the use of the CLT to establish that the test-set error term $\mathcal{R}_n(f_*) - \mathcal{R}(f_*)$ is $O_P(n^{-1/2})$. Identifying the remaining term as the excess risk $\mathcal{E}$ leads directly to Equation 4. $\qquad \square$

This analysis differs from Williamson et al. 2023, in which the assumptions made on the convergence rate of the ML model led to $\mathcal{E} = o_P(n^{-1/2})$. In contrast, we argue that these assumptions are not satisfied in practice, leading to an estimation error term mainly depending on $\mathcal{E}$.

*Remark* 4.4. From this leading error term, we can explain the estimation process for our target quantity, which primarily depends on the estimation of the model $f_\star$. Furthermore, the decomposition in (3) allows us to anchor empirical phenomena within a formal framework. While *stability* and the *Rashomon effect* often appear in the literature as loosely defined concepts regarding sampling and optimization inconsistencies, we explicitly map them to the concrete estimation ($\mathcal{E}_{\text{est}}$) and optimization ($\mathcal{E}_{\text{opt}}$) error terms, which are well-defined.

*Remark* 4.5. This proposition highlights the distinction between the two ensembling strategies:

- Importance-level ensembling (*sub-models*): this strategy aggregates the results of Equation 4 over individual models. By doing so, it may reduce the stochastic remainder $O_P(n^{-1/2})$, but not the excess risks $1/B \sum \mathcal{E}(f_{\text{b}})$, which equals the individual excess risk $\mathcal{E}(f_{\text{b}})$ for $\text{b} = 1, \cdots, B$ due to the linearity of the expectation. Thus, it does not affect the main bias.

- Model-level ensembling (*ensemble*): For any convex loss, Jensen's inequality ensures that the excess risk of this ensemble is reduced: $\mathcal{E}(f_{\text{ens}}) \leq 1/B \sum_{b=1}^{B} \mathcal{E}(f_{\text{b}})$. Moreover, if the loss is strictly convex, such as the quadratic loss, this inequality is strict just by assuming that the individual models differ.

These observations reveal a regime where measuring the importance of an ensemble is preferred. Specifically, in the bias-dominant regime where the convergence rate of the excess risk is slower than $n^{-1/2}$. In this regime, acting on the excess risk via model-level ensembling reduces the leading error term. Conversely, if the convergence rate is faster than $n^{-1/2}$, the tradeoff between variance reduction in the sub-models and bias reduction in the ensemble is not theoretically obvious, as the remainder term itself is impacted by the excess risk (see term $r_n$ in the proof A.1). Empirical results showing the respective contributions of the bias and variance terms as a function of $n$ are presented in Figure 3.

The results from Proposition 4.3 regarding the estimation of the Bayes risk provide the basis for deriving the estimation error of the *LOCO* measure. This relationship is summarized by the following theorem, which demonstrates that for complex models with potentially slow convergence rates, the importance estimation error is governed by the model's excess risk.

**Theorem 4.6** (Excess risk dominates importance estimation with *LOCO* ). *Denoting $\mathcal{E}^{-j} = \mathcal{R}(f_{\theta,\mathcal{D}_n}^{-j}) - \mathcal{R}(f_\star^{-j})$, the excess risk of the restricted model excluding feature $j$. The error in estimating the importance of a feature $j$ can be*

*decomposed as follows,*

$$\psi_n^{\text{loco}}(j) - \psi_\star^{\text{loco}}(j) = \mathcal{E}^{-j} - \mathcal{E} + O_P(n^{-1/2}) \quad (5)$$

From the above formulation, considering an ensembling strategy that reduces variance while leaving the bias unchanged, we obtain the following result.

**Proposition 4.7** (Impact of ensembling on importance estimation). *Consider an ensemble $f_{\text{ens}}$ of $B$ models with pairwise output correlation $\rho$. For LOCO importance with the MSE loss, let $\psi_\star$ be the true importance, and denote the estimated importances as $\psi_n^{\text{ens}} = \psi(f_{\text{ens}})$ and $\psi_n^{\text{sub}} = \frac{1}{B} \sum_{b=1}^{B} \psi(f_b)$. Then:*

$$\psi_n^{\text{ens}} - \psi_\star = (\psi_n^{\text{sub}} - \psi_\star) \left( \rho + \frac{1-\rho}{B} \right) + O_P(n^{-1/2})$$

*Proof sketch.* We analyze the difference in importance estimation error by comparing the excess risks governing Equation 5. Applying the bias-variance decomposition $\mathcal{E}(f) = \text{bias}(f)^2 + \text{Var}(f)$, we note that by linearity of expectation, $\mathbb{E}[f_{\text{ens}}] = \mathbb{E}[f_{\text{b}}]$, implying equal squared biases for any sub-model $f_{\text{b}}$: $\text{bias}(f_{\text{ens}})^2 = \text{bias}(f_{\text{b}})^2$. Thus, the bias terms cancel, leaving the difference to depend solely on variance. The final result follows by substituting the variance reduction for ensembling (Hastie et al., 2009): $\text{Var}(f_{\text{ens}}) = \left( \rho + \frac{1-\rho}{B} \right) \text{Var}(f_{\text{b}})$. We conclude by recalling that $\mathcal{E}_{\text{sub}} = \mathcal{E}_b$. $\square$

Crucially, this implies that when model diversity increases (lower $\rho$), the benefits of the *ensemble* strategy also increase. This can practically be achieved through ensembling techniques such as *bagging* or *voting*.

### 4.2. *SAGE*

The above results, obtained for *LOCO*, can be extended to the popular *SAGE* framework (Covert et al., 2020). Unlike *LOCO*, this approach is based on *marginalization*. For a subset of features $S$, instead of refitting a model on the complement $\bar{S}$, it marginalizes them out by averaging predictions, which gives the restricted model: $f^S(X_S) = \mathbb{E}[f(X) \mid X_S]$. Similarly, we denote $\mathcal{E}^S$, the excess risk of this restricted model. While theoretically grounded in the conditional expectation $\mathbb{E}[f(X) \mid X_S]$, which requires sampling $X_{\bar{S}}$ from the conditional distribution $p(X_{\bar{S}} \mid X_S)$ Covert et al. 2020 establishes that exact conditional sampling is intractable. Indeed, the above *marginalization* has to be computed for every subset $S \subset [d]$, with $d$ the dimension, leading to a combinatorial complexity that is prohibitive for high-dimensional data. Accordingly, the standard formulation approximates the restricted model via marginal sampling, where features in the complement $\bar{S}$ are drawn from $p(X_{\bar{S}})$. This marginal

sampling is typically implemented by permuting the feature values. *SAGE* then relies on summing value functions which are defined for a model $f$ and a subset of features $S$ as $v_f(S) := \mathbb{E}\left[\mathcal{L}(\mathbb{E}\left[f(X)\right], y)\right] - \mathbb{E}\left[\mathcal{L}(f^S(X_S), y)\right]$. The formula for computing *SAGE* values is given by,

$$\psi_\star^{\text{sage}}(j) = \frac{1}{d} \sum_{S \subseteq [d] \setminus j} \binom{d-1}{|S|}^{-1} \left[v_{f_\star}(S \cup \{j\}) - v_{f_\star}(S)\right],$$
(6)

where the sum is computed over all possible subsets $S$ of features, excluding $j$. The extension of the previous analysis to *SAGE*, however, requires additional assumptions.

**Assumption 4.8** (Support positivity). The input data $X \in [0,1]^p$ admits a density over $[0,1]^p$ bounded from above and below by strictly positive constants.

**Assumption 4.9** (Lipschitz continuity). The loss is $K$-Lipschitz-continuous, $||\mathcal{L}(\hat{f}, Y) - \mathcal{L}(f, Y)||_2 \leq K||\hat{f} - f||_{L^2}$

Assumption 4.8, which was introduced in Bénard et al. (2022), is necessary to extend the *loss consistency* from Assumption 4.1 to permuted inputs drawn from $p(X_{\bar{S}})$. However, this assumption is not required when considering the conditional version of *SAGE*, where $X_{\bar{S}}$ is sampled from the conditional distribution $P(X_{\bar{S}}|X_S)$. It should be noted that this is a strong assumption about the data distribution that limits the strength of feature dependencies, whereas Assumptions 4.1 and 4.2 only place constraints on the loss. In contrast, Assumption 4.9 is a standard regularity condition also made in Bottou & Bousquet (2007).

**Theorem 4.10** (Excess risk dominates estimation error with *SAGE*). *Under Assumptions 4.1, 4.2, 4.8 and 4.9*

$$\psi_n^{\text{sage}}(j) - \psi_\star^{\text{sage}}(j) = \frac{1}{d} \sum_{S \subseteq D \setminus j} \binom{d-1}{|S|}^{-1} \left(\mathcal{E}^{S \cup j} - \mathcal{E}^S\right) + O_P(n^{-1/2}) \quad (7)$$

Similar to *LOCO*, the error in estimating *SAGE* importance depends on the error in estimating the restricted models. However, these terms are summed over all possible feature subsets for *SAGE*.

These results for *LOCO* and *SAGE* highlight the role of the excess risk when estimating feature importance. This aspect has been overlooked in previous literature, which made strong assumptions about the convergence rate of ML models (Williamson et al., 2023). This in particular reveals a trade-off that motivates using ensembling at the model level, contrary to previous work that suggests aggregating importances of individual models (Donnelly et al., 2023).

### 4.3. *CFI*

We extend our analysis to Conditional Feature Importance (*CFI*), defined as a risk difference under input *perturbation*, $\psi_\star^{\text{cfi}}(j) = \mathcal{R}(f_\star(X^{\pi(j|-j)})) - \mathcal{R}(f_\star(X))$, where the $j^{th}$ feature in $X^{\pi(j|-j)}$ is sampled from the conditional distribution $P(X^j|X^{-j})$. Unlike LOCO or SAGE, this method applies the same model, $f_\star$ to both terms. Consequently, during the analysis of the estimation error, the quadratic excess risk terms cancel out. This leads to an error dependence on the model estimation that is linear, taking the form $O_P(\mathbb{E}[f_{\theta,\mathcal{D}_n} - f_\star])$. The linear dependence on the model error renders the two ensembling strategies theoretically equivalent for this term. This contrasts with *LOCO* and *SAGE*, where the excess risk is quadratic and can be directly reduced by model-level ensembling. Thus, improved predictive accuracy here primarily mitigates the second-order stochastic remainder (consistent with Remark 4.5). Detailed analysis and empirical results are provided in Appendix C.

### 4.4. PFI

Finally, we study Permutation Feature Importance (*PFI*), which corresponds to a risk difference under simple permutation, $\psi_\star^{\text{pfi}}(j) = \mathcal{R}(f_\star(X^{\pi(j)})) - \mathcal{R}(f_\star(X))$, where the $j^{th}$ feature in $X^{\pi(j)}$ is drawn from the marginal distribution $P(X^j)$ via permutation across samples. The outcome of the analysis is very similar to *CFI* but requires an additional assumption, support positivity (4.8), similarly to *SAGE*. Indeed, as with *CFI*, the same model $f^\star$ is applied to both $X^{\pi(j)}$ and $X$. As a result, for a model $f_\theta$, the excess risk terms $\mathcal{E}(f_\theta)$ that appear on both sides of the difference computed by *PFI* are the same. Consequently, they cancel out, leaving a linear dependence $O_P(\mathbb{E}[f_\theta - f^\star])$, similarly to *CFI*. However, unlike *CFI*, which preserves the data distribution through conditional sampling, *PFI* relies on marginal permutations that may produce inputs $X^{\pi(j)}$ outside the joint distribution's support. Consequently, the loss consistency of Assumption 4.1 does not automatically extend to these perturbed inputs. To ensure the model remains a valid risk minimizer on the permuted data, *PFI* requires the additional support positivity condition of Assumption 4.8, similar to the requirements for *SAGE*. Under this additional assumption, PFI behaves like CFI. These results are supported by empirical evidence, provided in Appendix D.

## 5. Experiments

The code to reproduce all experiments is publicly available.[1]

**Models** Our experiments focus on two model classes: Multi-Layer Perceptrons (MLP) and tree-based models. The

---

[1] https://github.com/jpaillard/ensemble_vim

MLP architecture consists of three hidden layers (64, 32, and 8 neurons). With more than 3,500 trainable parameters, it allows studying the overparametrized regime. Training is performed using the Adam optimiser with early stopping (patience of 10 epochs) for up to 500 epochs. The voting ensemble is obtained by randomising the weight values, using the Glorot & Bengio 2010 initialisation. For tree-based models, we use a decision tree regressor from *scikit-learn* with the squared-error splitting criterion and no limit on the maximum depth. Algorithmic stochasticity arises from the random set of features used to identify the optimal split at each node. We refer to the ensemble as a Random Forest (RF). For both RF and MLP, bagging ensembles are generated by training individual models on bootstrap samples of the data.

**Datasets** The datasets cover diverse non-linear relationships and varying levels of feature interactions. For all datasets, the number of features is fixed at $d = 20$. When the data-generating process involves fewer variables, we add spurious features unrelated to the outcome. The first dataset is *Friedman 1* (Friedman, 1991), where each feature follows a uniform distribution $X_j \sim \mathcal{U}(0, 1)$ and the response is defined by $Y = 10 \cdot \sin(\pi \cdot X_0 \cdot X_1) + 20 \cdot (X_2 - 0.5)^2 + 10 \cdot X_3 + 5 \cdot X_4$. The second is the *G-function* (Saltelli et al., 2004). The data-generating process is given by $Y = \prod_j (|4X_j - 2| + a_j)/(1 + a_j)$ with $a_j = j$ for $j \leq 5$ and $a_j = 100$ otherwise. Values of $a_j$ greater than 10 correspond to virtually insignificant features. In addition to the strong interactions of features in the generation of the response, the features have a correlation of $\rho = 0.3$. Lastly, we used the *Ishigami* function (Saltelli et al., 2004), $Y = \sin(X_0) + 7 \cdot \sin(X_1)^2 + 0.1 \cdot X_2^4 \cdot \sin(X_0)$. Here, the features are sampled from $\mathcal{U}(-\pi, \pi)$ with a correlation of $\rho = 0.3$. Gaussian noise is added to each response to achieve a signal-to-noise ratio of one. Notably, for all three datasets in the absence of noise and with large sample sizes (e.g., $n = 3 \times 10^4$), both RF and MLP achieve an $R^2$ score almost equal to 1. This allows us to focus on the estimation and optimization errors effectively mitigated by ensembling while ignoring the approximation error (see Figure 1).

The first results on the *Friedman 1* dataset, presented in Figure 2, demonstrate that for both *LOCO* and *SAGE*, ensembling at the model level consistently improves feature importance estimation across both RF and MLP architectures. As illustrated in the top row of Figure 2, this strategy leads to a lower MSE, where the ground truth is defined as the asymptotic importance value estimated using $n = 10^5$. Reducing this metric is critical for feature ranking, as larger estimation errors can lead to incorrect orderings. Moreover, model-level ensembling improves classification metrics for identifying relevant features, yielding higher ROC AUC scores. This advantage holds whether the classification tar-

get is the Markov blanket (*LOCO*) or the set of features functionally linked to the outcome (*SAGE*). The ground-truth binary labels designate any feature with non-zero asymptotic importance as relevant. This empirical finding is particularly significant because it demonstrates that reductions in estimation error effectively translate into improved selection performance. This outcome is not trivial, as lower MSE does not inherently guarantee better feature classification.

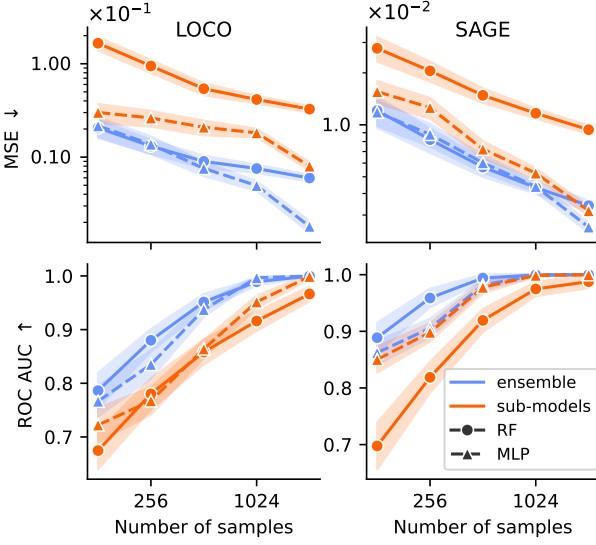

*Figure 2.* **Model-level ensembling reduces feature importance estimation error and improves selection.** Importance is measured directly on the bagging ensemble (blue) versus averaging individual sub-model importances (orange) on Friedman 1. Model-level ensembling yields both lower MSE, $(\psi_n - \psi_\star)^2$ and higher ROC AUC, indicating more accurate variable ranking and superior feature selection. These gains hold for both *LOCO* and *SAGE* across Random Forest (solid) and MLP (dashed) architectures. Error bars represent one standard deviation over 100 random seeds.

Figures 6-13 in appendix provide extended results on these performance gains across the full suite of benchmarks, isolating the sources of estimation error: sampling instability versus algorithmic stochasticity. For *LOCO* and *SAGE*, we observe that the performance gap between the ensemble and sub-model strategies is more pronounced under bagging than voting. When focusing on algorithmic stochasticity via voting, the gap is notably larger for MLPs than for Random Forests. This distinction reflects neural networks' greater sensitivity to weight initialization compared to the randomized feature selection used in tree splitting. Crucially, model-level ensembling reduces estimation error for both support and null features; the former is essential for accurately ranking important variables, while the latter matters for limiting false discoveries in high-dimensional settings. By contrast, and consistent with our theoretical findings, *CFI* shows negligible differences between strategies.

The second experiment provides a deeper understanding of the performance gains from model-level ensembling, consistent with the theoretical discussion in Remark 4.5. We present in Figure 3 a decomposition of the MSE into squared bias and variance across varying sample sizes, estimated over 100 random data draws from the Friedman 1 dataset. This decomposition reveals the role of the bias term in the total error, which is effectively reduced by model-level ensembling. Consequently, while the sub-models strategy can be more efficient at reducing variance, it remains hampered by significant bias, leading to a higher overall MSE.

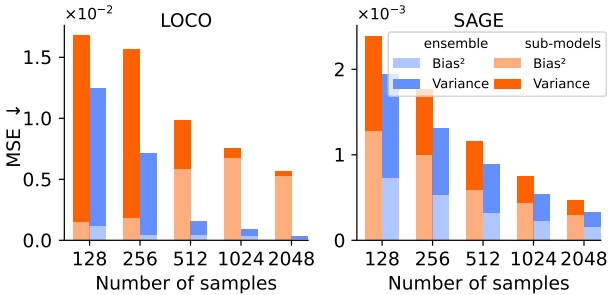

*Figure 3.* **Bias-variance decomposition of feature importance estimation error.** The bar plots show the contributions of the squared bias (dark shades) and the variance (light shades) to the MSE for both the ensemble (blue) and sub-models (orange) strategies. Estimation error corresponds to the MSE, $(\psi_n - \psi_\star)^2$ (Equation 5 and 7). These results were obtained using an MLP model on the Friedman 1 dataset.

The benchmark presented in Figure 4 demonstrates that across all three datasets, the *ensemble* strategy consistently reduces MSE and increases the ROC AUC for feature selection. In this experiment, the sample size is fixed at $n = 512$, and we focus on *LOCO* importance. The improvements in MSE and ROC AUC can be attributed to the predictive performance gains obtained by ensembling, as reflected by the systematically higher $R^2$ scores reported in the first column. This observation aligns with the theoretical result in Theorem 4.6, which explicitly links the reduction of excess risk to a reduction in feature importance estimation error.

These findings generalize beyond the $d = 20$ benchmarks presented here: Appendix H extends the comparison to a high-dimensional setting with $d = 100$ features and an over-parameterized MLP, where the ensemble strategy maintains its advantage in both importance MSE and support recovery. Appendix I further replicates this benchmark using TabICL (Qu et al., 2025), a tabular foundation model, showing consistent improvements in importance MSE. Beyond MSE and ROC AUC, Appendix J evaluates ranking stability via Spearman's rank correlation across cross-validation folds, confirming that the ensemble strategy also produces more consistent importance rankings. We further validate these findings on a breast cancer gene expression dataset with

established ground-truth driver genes, where the ensemble strategy achieves higher recovery of known cancer drivers (Appendix G).

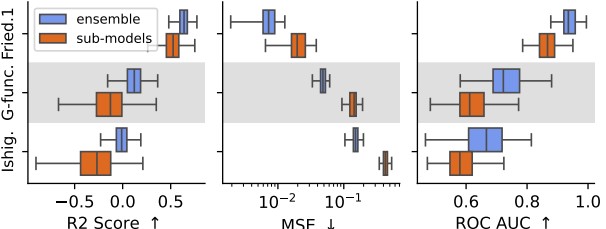

*Figure 4.* **Ensemble-level importance improves estimation across diverse datasets.** Measuring *LOCO* importance directly on the bagging ensemble (blue) consistently outperforms the average of sub-model scores (orange). Higher $R^2$ scores demonstrate the ensemble's superior predictive performance. This improved prediction translates into more accurate numerical estimates for ranking (lower MSE) and more reliable feature selection (higher ROC AUC). These gains are robust across the Friedman 1, G-function, and Ishigami datasets (rows of the plots), which present varying non-linearities and levels of feature interactions. For all datasets, the number of samples was set to n=512 and the estimator is a MLP. Box plots represent results across 100 random seeds.

Lastly, we demonstrate the clinical utility of our framework through a real-world application: identifying the proteomic signatures associated with Body Mass Index (BMI) using the UK Biobank (UKBB, Sudlow et al. 2015). While BMI is a standard proxy for obesity—a primary risk factor for type 2 diabetes and cardiovascular disease—it is a crude measure failing to account for body composition. Its correlation with body fat holds primarily at the population level, often misclassifying individuals with high muscle-to-fat ratios (e.g., athletes) as obese. Furthermore, BMI frequently fails to capture meaningful metabolic improvements following lifestyle interventions, such as changes in body composition or the loss of visceral fat, when total body weight remains stable (Prentice & Jebb, 2001). In contrast, blood-measured proteins can provide a more granular reflection of an individual's internal physiological state, capturing heterogeneous metabolic phenotypes and dietary patterns that anthropometric metrics overlook (Watanabe et al., 2023). We predicted BMI from a panel of $2,922$ plasma proteins measured via the Olink platform in a cohort of $n = 46,382$ UKBB participants. The prediction pipeline consisted of a feature selection step to retain the 50 variables with the strongest univariate linear associations. This was followed by an ensemble of 10 *LightGBM* models, with varying hyperparameters (learning rate ranging from 0.01 to 1 and max depth ranging from 5 to 30), each trained on a bootstrap sample of the data. The model was implemented using the *scikit-learn* library (Pedregosa et al., 2011). The ensemble achieved an $R^2$ score of $0.62 \pm 0.001$ via 5-fold cross-validation, which outperforms individual models (mean $R^2 = 0.54$) and is comparable to recent multiomic BMI studies on inde-

pendent cohorts (Watanabe et al., 2023). Figure 5 presents importance scores measured with *LOCO*, a measure adapted to the inherent multicollinearity of such a large proteomic panel. Additional results for *SAGE* and *CFI* are reported in Appendix F.

The feature importance rankings revealed practical differences between ensembling strategies across the UKBB cohort. Both strategies identified established metabolic markers, including FABP4 (Fatty Acid Binding Protein 4), which is strongly associated with adiposity (Watanabe et al., 2023), LEP (Leptin), an adipocyte-derived hormone regulating long-term energy balance and satiety, and ADM (Adrenomedullin), an inhibitor of endothelial insulin signaling that increases in concentration during obesity (Cho et al., 2025). Crucially, model-level ensembling improved the estimation of additional proteins with clear biological underpinnings. These include IGFBP-1 and -2 (Insulin-like Growth Factor Binding Proteins), which are inversely related to insulin levels and fat mass. Notably, IGFBP-2 is recognized for its specific role in protecting against the development of insulin resistance and obesity (Wheatcroft et al., 2007). In contrast, the importance estimated with *sub-models* showed error bars (one standard deviation across folds) that largely overlapped with 0.

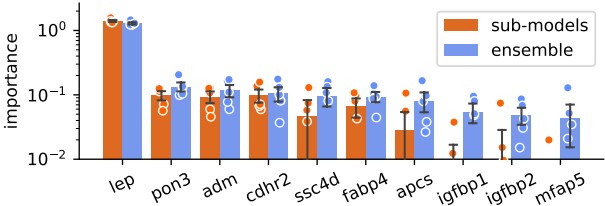

*Figure 5.* **Identification of proteomic signatures for BMI in the UK Biobank.** Feature importance ranking of the top 10 proteins identified with *LOCO* for the prediction of BMI from 2,922 proteins measured in plasma using the Olink platform ($n = 46,382$ participants). The predictive model was an ensemble comprising 10 *LightGBM* models. Error bars indicate one standard deviation estimated via 5-fold cross-validation. For each fold, the ensemble importance score and the mean importance across individual models are represented by blue and orange circles, respectively.

## 6. Discussion

Our analysis shifts the perspective on improving feature importance accuracy. Under assumptions reflecting the slow convergence rates of state-of-the-art models, we find that the model's estimation error is the main driver of importance inaccuracy. This suggests clear guidelines that were previously overlooked: ensembling at the model level improves over aggregating individual models' importances. We empirically validate these theoretical findings across diverse benchmarks and real-world data. These results also explain the poor performance of *LOCO* when used without

ensembling, which was documented in the recent literature (Paillard et al., 2025). Importantly, because importance methods have distinct mathematical formulations and rely on different assumptions, there is no single general theorem that covers all methods; each requires a specific theoretical treatment. Our analysis reveals that the benefits of model-level ensembling depend on the method. For risk-difference methods such as *LOCO* and *SAGE*, the importance estimation error depends quadratically on the model's excess risk, leading to substantial gains from ensembling. In contrast, for *CFI* and *PFI*, the same model is evaluated on both original and perturbed inputs, so that excess risk terms cancel, leaving only a linear dependence. While model-level ensembling empirically consistently improves importance estimation across all methods, the magnitude of these gains depends heavily on the importance measure. This nuance would not emerge without formal analysis.

**Limitations and future work** We focused our analysis on methods targeting well-defined theoretical indices to enable rigorous error decomposition. This scope limits our work by excluding heuristic methods (e.g., permutation importance, gradient-based attribution) that lack population-level estimands, and it restricts empirical validation to tabular data. Unlike tabular features, images or time series lack the semantic consistency required to define valid population-level indices without prior representation learning.

**Acknowledgements** This research has received funding from the H2020 Research Infrastructures Grant EBRAIN-Health 101058516 and the VITE ANR23-CE23-0016 and PEPR Sante numérique, Brain health Trajectories ANR-22-PESN-0012 projects. This research has been conducted using the UK Biobank Resource under Application Number 44257.

## Impact Statement

While motivated by scientific discovery in biomedical research, improving the reliability of explanation methods is also critical for the regulation of high-risk Artificial Intelligence (AI). Reliable explanations can play a pivotal role in globally emerging legislative frameworks; for instance, Article 86 of the EU AI Act explicitly grants individuals a "Right to Explanation" regarding automated decisions. Ensuring the robustness of explanations is therefore a necessary step to facilitate the implementation of such regulation.

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

# A. Proofs

## A.1. Proof of Proposition 4.3

**Assumption: *Loss consistency***    $f_{\theta,\mathcal{D}_n}$ is a minimizer of the risk: $||\mathcal{L}(f_{\theta,\mathcal{D}_n}(X),Y) - \mathcal{L}(f_\star(X),Y)||_2 = o_P(1)$

**Assumption: *Finite variance of the loss***    The loss function, evaluated at the risk minimizer $f_\star$, has a finite variance with respect to the data distribution:
$\mathrm{Var}\left(\mathcal{L}(f_\star(X),Y)\right) < \infty$

**Proposition: Bayes risk estimation error**    Under Assumptions 4.1 and 4.2, we have that

$$\mathcal{R}_n\left(f_{\theta,\mathcal{D}_n}\right) - \mathcal{R}(f_\star) = \mathcal{E} + O_P(n^{-1/2}). \tag{8}$$

*Proof.* **(1) decomposition**
We start by using a decomposition similar to Equation 2, leading to:

$$\mathcal{R}_n\left(f_{\theta,\mathcal{D}_n}\right) - \mathcal{R}(f_\star) = \underbrace{[\mathcal{R}_n(f_\star) - \mathcal{R}(f_\star)]}_{A} - \underbrace{[\mathcal{R}(f_{\theta,\mathcal{D}_n}) - \mathcal{R}(f_\star)]}_{\mathcal{E}} + r_n$$

We will also treat separately the terms $A, B, r_n$.

**(2) term $A$**
We start by studying $A = \mathcal{R}_n(f_\star) - \mathcal{R}(f_\star)$. It is important to note that $f_\star$ is fixed and does not depend on the training data. We will therefore directly apply the central limit theorem, using that

$$\mathcal{R}_n(f_\star) - \mathcal{R}(f_\star) = \frac{1}{n}\sum_{i=1}^{n}\left(\mathcal{L}(f_\star(X_i),Y_i) - \mathbb{E}\left[\mathcal{L}(f_\star(X),Y)\right]\right)$$

$(X_i, Y_i)$ being independent and identically distributed. Then the assumption 4.2, on the *finite variance of the loss*, that we denote $\sigma^2 = \mathrm{Var}\left(\mathcal{L}(f_\star(X),Y)\right) < \infty$, allows us to use the central limit theorem, which gives $\sqrt{n}\left(\mathcal{R}_n(f_\star) - \mathcal{R}(f_\star)\right) \to \mathcal{N}(0,\sigma^2)$.

**Note**: Using the assumption that the loss $\mathcal{L}$ is bounded by $l_\infty$, Hoeffding's inequality could also be used to bound the risk differences that with probability greater than $1 - \delta$,

$$\mathcal{R}_n(f_\star) - \mathcal{R}(f_\star) \le \frac{l_\infty}{\sqrt{2n}}\sqrt{\log\frac{1}{\delta}}, \tag{9}$$

**(3) term $r_n$**

This proof for this term relies on *Lemma 1* from Kennedy 2024, which state that for a function $f_{\theta,\mathcal{D}_n}$ fitted on a separate train set,

$$r_n = [\mathcal{R}_n(f_{\theta,\mathcal{D}_n}) - \mathcal{R}(f_{\theta,\mathcal{D}_n})] - [\mathcal{R}_n(f_\star) - \mathcal{R}(f_\star)]$$
$$= O_P\left(\frac{||\mathcal{L}(f_{\theta,\mathcal{D}_n}(X),Y) - \mathcal{L}(f_\star(X),Y)||_{\mathcal{F}}}{\sqrt{n}}\right)$$

The proof of this lemma relies on showing that $r_n$ is zero-mean and has a variance bounded by $||\mathcal{L}(f_{\theta,\mathcal{D}_n},Y) - \mathcal{L}(f_\star,Y)||_{\mathcal{F}}^2/n$, which then allows using Chebyshev's inequality. Then, using assumption 4.1 on the *loss consistency*, we have that $||\mathcal{L}(f_{\theta,\mathcal{D}_n}(X),Y) - \mathcal{L}(f_\star(X),Y)||_{\mathcal{F}} = o_P(1)$ and consequently $r_n = o_P(n^{-1/2})$

**(4) conclusion**
The remaining term $B = \mathcal{R}(f_{\theta,\mathcal{D}_n}) - \mathcal{R}(f_\star)$ corresponds to the excess risk, that we denote $\mathcal{E}$. Combining all three terms gives

$$\mathcal{R}_n\left(f_{\theta,\mathcal{D}_n}\right) - \mathcal{R}(f_\star) = \mathcal{E} + O_P(n^{-1/2}),$$

which completes the proof. $\qquad\square$

### A.2. Proof of Theorem 4.6

**Theorem: Excess risk dominates importance estimation with *LOCO*** Denoting $\mathcal{E}^{-j} = \mathcal{R}(f_{\theta,\mathcal{D}_n}^{-j}) - \mathcal{R}(f_\star^{-j})$ the excess risk in estimating $f_\star^{-j}$. The error in estimating the importance of a feature $j$ can be decomposed as follow,

$$\psi_n^{\mathrm{loco}}(j) - \psi_\star^{\mathrm{loco}}(j) = \mathcal{E}^{-j} - \mathcal{E} + O_P(n^{-1/2})$$

**(1) Decomposition** We start by writting down the expression of the feature importance estimate for the feature $j$, $\psi_n^{\mathrm{loco}}(j)$ and its target quantity, $\psi_\star^{\mathrm{loco}}(j)$:

$$\psi_n^{\mathrm{loco}}(j) - \psi_\star^{\mathrm{loco}}(j) = \mathcal{R}_n\left(f_{\theta,\mathcal{D}_n}^{-j}\right) - \mathcal{R}_n\left(f_{\theta,\mathcal{D}_n}\right) - \mathcal{R}\left(f_\star^{-j}\right) + \mathcal{R}(f_\star) \qquad (10)$$

**(2) Application of Proposition 4.3** We can then apply Proposition 4.3, which gives $\mathcal{R}_n\left(f_{\theta,\mathcal{D}_n}^{-j}\right) - \mathcal{R}\left(f_{-j}\right) = \mathcal{E}^{-j} + O_P(n^{-1/2})$. Where $\mathcal{E}^{-j} = \mathcal{R}\left(f_{\theta,\mathcal{D}_n}^{-j}\right) - \mathcal{R}(f_{-j})$ is the excess risk of the model $f_{\theta,\mathcal{D}_n}^{-j}$. To the same extent, we have $\mathcal{R}_n\left(f_{\theta,\mathcal{D}_n}\right) - \mathcal{R}(f_\star) = \mathcal{E} + O_P(n^{-1/2})$. And consequently,

$$\psi_n^{\mathrm{loco}}(j) - \psi_\star^{\mathrm{loco}}(j) = \mathcal{E}^{-j} - \mathcal{E} + O_P(n^{-1/2})$$

which completes the proof.

### A.3. Proof of Proposition 4.7

**Impact of bagging on importance estimation** Consider an ensemble $f_{\mathrm{ens}}$ of $B$ models with pairwise output correlation $\rho$. For any feature, let $\psi_\star$ be the true importance, and denote the estimated importances as $\psi_{\mathrm{ens}} = \psi(f_{\mathrm{ens}})$ and $\psi_{\mathrm{sub}} = \frac{1}{B}\sum_{b=1}^B \psi(f_b)$. Then, we have:

$$\psi_{\mathrm{ens}} - \psi_\star = (\psi_{\mathrm{sub}} - \psi_\star)\left(\rho + \frac{1-\rho}{B}\right) + O_P(n^{-1/2})$$

**(1) Notations** We denote the error in estimating the importance of feature $j$, by using the model-level ensembling. We explicit the dependence of the feature importance estimate on the model through its first argument,

$$\begin{aligned} \Delta_{\mathrm{ens}} &= \psi_n^{\mathrm{loco}}(\hat{f}_{\mathrm{ens}}, j) - \psi_\star^{\mathrm{loco}}(f_\star, j) \\ &= \mathcal{E}(\hat{f}_{\mathrm{ens}}^{-j}) - \mathcal{E}(\hat{f}_{\mathrm{ens}}) + O_P(n^{-1/2}) \end{aligned}$$

Where $\hat{f}_{\mathrm{ens}}$ is the notation used for the estimated ensemble model, such that $\hat{f}_{\mathrm{ens}}(\cdot) = \frac{1}{B}\sum_{b=1}^B \hat{f}_b(\cdot)$. Here the notation $\hat{f}_b$ indicates the randomization of the individual models $f_1, \cdots, f_B$ with regard to the bootstrap sample of the training data. Then we denote

$$\begin{aligned} \Delta_{sub} &= \frac{1}{B}\sum_{b=1}^B \psi_n^{\mathrm{loco}}(\hat{f}_b, j) - \psi_\star^{\mathrm{loco}}(f_\star, j) \\ &= \frac{1}{B}\sum_{b=1}^B \left(\mathcal{E}(\hat{f}_b^{-j}) - \mathcal{E}(\hat{f}_b)\right) + O_P(n^{-1/2}) \end{aligned}$$

**(2) bias-variance decomposition** Then, using the bias-variance decomposition of the error, we have $\mathcal{E}(\hat{f}_b) = \text{Bias}(\hat{f}_b)^2 +$ $\text{Var}(\hat{f}_b)$. Since each sub-model is trained on a bootstrap sample of the training data, they are identically distributed and share the same bias term, $\text{Bias}(\hat{f}_{\text{ens}}) = \text{Bias}(\hat{f}_b)$. On the contrary, bagging reduces the variance. Asssuming that the models have a correlation of $\rho$, we have that $\text{Var}(\hat{f}_{\text{ens}}) = \left(\rho + \frac{1-\rho}{B}\right) \text{Var}(\hat{f})$ (Hastie et al., 2009). Where $\text{Var}(\hat{f})$ denotes the common variance of any individual model $\hat{f}_b$.

**(3) Conclusion** Assembling the two previous points together, we have,

$$\begin{aligned}
\Delta_{\text{ens}} &= \mathcal{E}(\hat{f}_{\text{ens}}^{-j}) + \mathcal{E}(\hat{f}_{\text{ens}}) + O_P(n^{-1/2}) \\
&= \left(\rho + \frac{1-\rho}{B}\right)\left(\text{Var}(\hat{f}^{-j}) - \text{Var}(\hat{f})\right) + O_P(n^{-1/2}) \\
&= \Delta_{sub} \cdot \left(1 + \frac{1-\rho}{B}\right) + O_P(n^{-1/2})
\end{aligned}$$

## A.4. Proof of Theorem 4.10

**Support positivity** The input data $X \in [0,1]^p$ admits a density over $[0,1]^p$ bounded from above and below by stricly positive constants.

**Lipschitz continuity** The loss is $K$-Lipschitz-continuous, $||\mathcal{L}(\hat{f}, Y) - \mathcal{L}(f, Y)||_2 \leq K||\hat{f} - f||_2$

**Proposition: Excess risk dominates importance estimation with *SAGE*** Under Assumptions 4.1, 4.2 and 4.8,

$$\begin{aligned}
\psi_n^{\text{sage}}(j) - \psi_\star^{\text{sage}}(j) = \frac{1}{d} \sum_{S \subseteq D \setminus j} \binom{d-1}{|S|}^{-1} \left(\mathcal{E}^{S \cup j} - \mathcal{E}^S\right) \\
+ O_P(n^{-1/2})
\end{aligned} \tag{11}$$

**(1) Notations** We recall that the restricted model is given by $f_\star^S(X^S) = \mathbb{E}\left[f_\star(X) \mid X^S\right]$ and that instead of sampling variables of the complement $\bar{S}$ from the conditional distribution, they are sampled from the marginal distribution $p(X^{\bar{S}}) = \prod_{j \in \bar{S}} p(X^j)$ which is achieved by randomly permuting $X^{\bar{S}}$

$$v_{f_\star}(S) := \mathbb{E}\left[\mathcal{L}(\mathbb{E}\left[f_\star(X)\right], Y)\right] - \mathbb{E}\left[\mathcal{L}(f_\star^S(X^S), Y)\right].$$

To assign the importance of a feature $X^j$ with respect to a subset of other features $S \subset [d]$, *SAGE* compares the combinations of value functions across all possible sets. For clarity, we will omit the upper script in this proof, denoting $\psi(j) = \psi^{\text{sage}}(j)$ as it is unambiguous in this context.

**(2) SAGE estimation error.** The estimation error of SAGE satisfies

$$\psi_j^* - \hat{\psi}_n(j) = \frac{1}{d} \sum_{S \subseteq D \setminus \{j\}} \binom{d-1}{|S|}^{-1} \left[v_{f_\star}(S \cup \{j\}) - v_{f_\star}(S) - \left(\hat{v}_f(S \cup \{j\}) - \hat{v}_f(S)\right)\right].$$

Thus, we need to study $\hat{v}_f(S) - v_{f_\star}(S)$. Note that the marginalization in $f_\star^S(X^S)$ is done in practice by averaging across $n_{\text{cal}}$ permutations. To take into account this additional source of approximation, we use the notation $\hat{f}_{\text{cal}}^S(X_i) = \frac{1}{n_{\text{cal}}} \sum_\ell \hat{f}(X_i^S, X_\ell^{\bar{S}})$, were $X_\ell^{\bar{S}}$ corresponds to a permutation of the features in $\bar{S}$ whereas the features in $S$, denoted $X_i^S$ are unchanged.

**(3) Decomposition.** We write

$$\hat{v}_f(S) - v_{f_\star}(S) = \frac{1}{n_{\text{test}}} \sum_i \mathcal{L}\left(\hat{f}^S_{\text{cal}}(X_i), Y_i\right) - \mathbb{E}\left[\mathcal{L}(f^S_\star(X^S), Y)\right]$$

$$= \frac{1}{n_{\text{test}}} \sum_i \left( \mathcal{L}(f^S_\star(X^S_i), Y_i) - \mathbb{E}\left[\mathcal{L}(f^S_\star(X^S), Y)\right] \right)$$

$$+ \mathbb{E}\left[\mathcal{L}\left(\hat{f}^S_{\text{cal}}(X), Y\right)\right] - \mathbb{E}\left[\mathcal{L}(f^S_\star(X^S), Y)\right] + r_n$$

$$= A + B + r_n.$$

**(4) Term $A$.** The assumption on the finite variance of the loss allows using the CLT. We then obtain:

$$A = \frac{1}{n_{\text{test}}} \sum_i \left( \mathcal{L}(f^S_\star(X^S_i), Y_i) - \mathbb{E}\left[\mathcal{L}(f^S_\star(X^S), Y)\right] \right) = O_p(n^{-1/2}).$$

**(5) Remainder term.** Using Lemma 1 of Kennedy (2020),we obtain,

$$r_n = \frac{1}{n_{\text{test}}} \sum_i \left( \mathcal{L}\left(\hat{f}^S_{\text{cal}}(X_i), Y_i\right) - \mathcal{L}(f^S_\star(X^S_i), Y_i) \right)$$

$$- \mathbb{E}\left[ \left( \mathcal{L}\left(\hat{f}^S_{\text{cal}}(X), Y\right) - \mathcal{L}(f^S_\star(X^S), y) \right) \right]$$

$$= O_p\left( \frac{\left\| \mathcal{L}\left(\hat{f}^S_{\text{cal}}(X), Y\right) - \mathcal{L}\left(f^S_\star(X^S), Y\right) \right\|}{\sqrt{n}} \right).$$

**Lipschitz control.** Since the model $f^S_{\text{cal}}$ predicts from permuted inputs, the assumption of loss consistency does not allow us to conclude on the convergence of the numerator. To prove that it indeed converges to 0, we split the error into a first part capturing the estimation of the marginal distribution from $n_{\text{cal}}$ permutation and a second part related to the model's error. We use the notation $\frac{1}{n_{\text{cal}}} \sum_{\ell=1}^{n_{\text{cal}}} f^{\text{cal}}_\star(X_\ell)$ to indicate that we average predictions over $n_{\text{cal}}$ permutations of $X^{\bar{S}}$, leaving $X^S$ unchanged. Indeed, note that using the decomposition and the Lipschitz assumption, we have that

$$\left\| \mathcal{L}\left(\hat{f}^S_{\text{cal}}, Y\right) - \mathcal{L}(f^S_\star(X^S), Y) \right\|$$

$$\leq \left\| \mathcal{L}\left(\hat{f}^S_{\text{cal}}, Y\right) - \mathcal{L}\left(\frac{1}{n_{\text{cal}}} \sum_\ell f^{\text{cal}}_\star(X_\ell), Y\right) \right\| + \left\| \mathcal{L}\left(\frac{1}{n_{\text{cal}}} \sum_\ell f^{\text{cal}}_\star(X_\ell), Y\right) - \mathcal{L}(f^S_\star(X^S), Y) \right\|$$

$$\leq \left\| \mathcal{L}\left(\hat{f}^S_{\text{cal}}, Y\right) - \mathcal{L}\left(\frac{1}{n_{\text{cal}}} \sum_\ell f^{\text{cal}}_\star(X_\ell), Y\right) \right\| + K \left\| \frac{1}{n_{\text{cal}}} \sum_\ell f^{\text{cal}}_\star(X_\ell) - f^S_\star(X^S) \right\|.$$

For the second term we use the CLT; for the first term, we use Assumption 4.1, which extends to $\hat{f}^S_{\text{cal}}$ thanks to Assumption 4.8 on the support positivity. Hence $r_n = o_p(n^{-1/2})$.

**(6) Term $B$.** We treat separately the marginalization error arising from averaging over the $n_{\text{cal}}$ permutations from the error induced by the estimation of the model. To do so, we decompose the errors:

$$B = \mathbb{E}\left[\mathcal{L}\left(\hat{f}^S_{\text{cal}}(X), y\right)\right] - \mathbb{E}\left[\mathcal{L}\left(\mathbb{E}[f_\star(X^{-S}) \mid X^S], y\right)\right]$$

$$= \left( \mathbb{E}\left[\mathcal{L}\left(\hat{f}^S_{\text{cal}}(x), y\right)\right] - \mathbb{E}\left[\mathcal{L}\left(\mathbb{E}[\hat{f}(X^{-S}) \mid X^S], y\right)\right] \right)$$

$$+ \left( \mathbb{E}\left[\mathcal{L}\left(\mathbb{E}[\hat{f}(X^{-S}) \mid X^S], y\right)\right] - \mathbb{E}\left[\mathcal{L}\left(\mathbb{E}[f_\star(X^{-S}) \mid X^S], y\right)\right] \right)$$

$$= B^1 + B^2.$$

For the first term, we use the Lipschitz assumption and the CLT to have that

$$B^1 \leq K\mathbb{E}\left[\left\| \hat{f}^S_{\text{cal}}(x) - \mathbb{E}[\hat{f}(X^{-S}) \mid X^S] \right\|\right] = O_p(n^{-1/2}).$$

**(7) Conclusion.** We conclude that

$$\hat{v}_f(S) - v_{f_\star}(S) = \underbrace{\mathbb{E}\Big[\mathcal{L}\Big(\mathbb{E}[\hat{f}(X^{-S}) \mid X^S], y\Big)\Big] - \mathbb{E}\big[\mathcal{L}\big(\mathbb{E}[f_\star(X^{-S}) \mid X^S], y\big)\big]}_{\mathcal{E}^S} + O_p(n^{-1/2}).$$

Similarly, we have

$$\hat{v}_f(S \cup \{j\}) - v_{f_\star}(S \cup \{j\}) = \mathcal{E}^{S\cup\{j\}} + O_p(n^{-1/2})$$

## B. Extended experiments

The extended experiments provide a comprehensive analysis of the results introduced in section 5, offering a detailed comparison of the *ensemble* versus *sub-models* strategies across a broad range of experimental conditions for *LOCO* (Figure 6) and *SAGE* (Figure 7). *CFI* (Figure 12) is left for Section E.1. We specifically isolate and compare two primary sources of estimation error: training sample instability and algorithmic stochasticity, examining the corresponding mitigation strategies—bagging and voting, respectively. While the *ensemble* approach consistently reduces estimation error, our results indicate that its benefits are particularly pronounced when mitigating sampling instability. Furthermore, we separate the results to analyze the estimation error for features within the true support versus null features (not in the support). This distinction reveals that while the MSE generally decreases at a faster rate for null features, the performance gap between the two strategies is often wider in this context. Finally, we report these findings across all three benchmarks: Friedman 1, Ishigami and G-function.

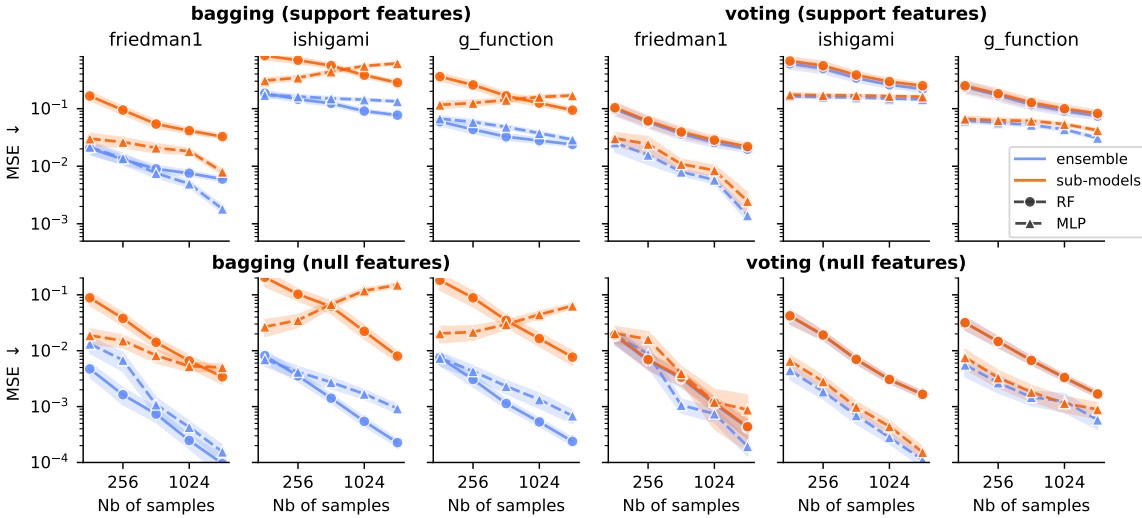

*Figure 6.* **Model-level ensembling consistently reduces *LOCO* importance estimation error.** Mean Squared Error (MSE) of *LOCO* feature importance estimates as a function of sample size ($n$) for Random Forest (RF) and Multi-Layer Perceptron (MLP) architectures. The plots compare two estimation strategies: ensemble (blue), where importance is derived from the aggregated model, and sub-models (orange), where importance scores from individual models are averaged. Top row: MSE for features within the true support. Bottom row: MSE for null features. Columns display results for the three benchmark datasets: Friedman 1, Ishigami, and G-function. The left panels correspond to bagging (ensembling over bootstrap training samples), while the right panels correspond to voting (ensembling over random initializations). The ensemble strategy consistently yields lower MSE, particularly for support features, validating the theoretical reduction in bias. Results are averaged over 100 independent trials from each data-generating process; error bars indicate 95% bootstrap confidence intervals of the mean.

Complementing the estimation error analysis, we evaluate the feature selection capabilities of each strategy using the Area Under the ROC Curve (ROC AUC). In this context, feature selection is framed as a binary classification task: distinguishing relevant features from null features based on their estimated importance scores. We define a feature as relevant if its asymptotic importance (computed at $n = 10^5$) is strictly positive; this criterion captures both the Markov blanket (for LOCO) and all functionally linked features (for SAGE). We report the mean ROC AUC and its standard deviation across 100 random seeds. The extended results, presented in Figure 8 (*LOCO*) and Figure 9 (*SAGE*), compare the *ensemble*

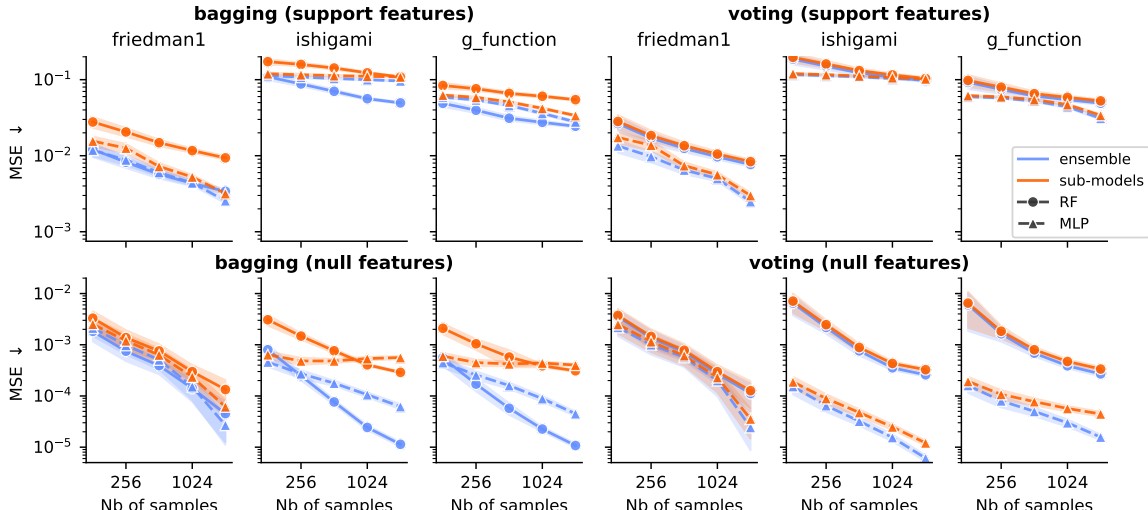

*Figure 7.* **Model-level ensembling consistently reduces *SAGE* importance estimation error.** Mean Squared Error (MSE) of *SAGE* feature importance estimates as a function of sample size ($n$) for Random Forest (RF) and Multi-Layer Perceptron (MLP) architectures. The plots compare two estimation strategies: ensemble (blue), where importance is derived from the aggregated model, and sub-models (orange), where importance scores from individual models are averaged. Top row: MSE for features within the true support. Bottom row: MSE for null features. Columns display results for the three benchmark datasets: Friedman 1, Ishigami, and G-function. The left panels correspond to bagging (ensembling over bootstrap training samples), while the right panels correspond to voting (ensembling over random initializations). The ensemble strategy consistently yields lower MSE, particularly for support features, validating the theoretical reduction in bias. Results are averaged over 100 random seeds; error bars represent the standard deviation of the ROC AUC.

and *sub-models* strategies across both bagging and voting regimes. These plots demonstrate that model-level ensembling generally yields higher ROC AUC scores, particularly in low-sample regimes, indicating a superior ability to distinguish relevant from spurious features.

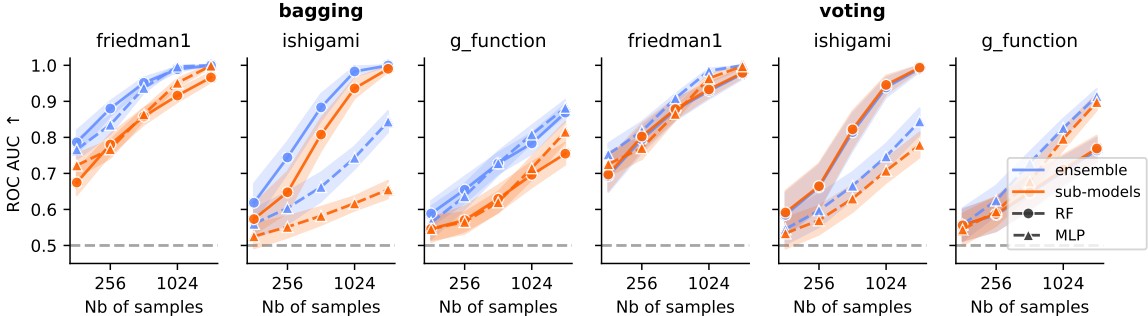

*Figure 8.* **Model-level ensembling improves feature selection performance with *LOCO*.** Area Under the ROC Curve (ROC AUC) for feature selection as a function of sample size ($n$). The ROC curves evaluate the ability to classify features as relevant versus irrelevant, where ground truth relevance is defined by having a non-zero asymptotic importance value (computed at $n = 10^5$). The estimated feature importance score serves as the decision function for the ROC. The plots compare the ensemble strategy (blue) against the sub-models strategy (orange) for RF and MLP architectures across the three benchmark datasets (Friedman 1, Ishigami, G-function). Left panels: Results using bagging (bootstrap training samples). Right panels: Results using voting (random initializations). Higher AUC values indicate that the method assigns higher importance scores to relevant features compared to null features. Results are averaged over 100 random seeds; error bars represent the standard deviation of the ROC AUC.

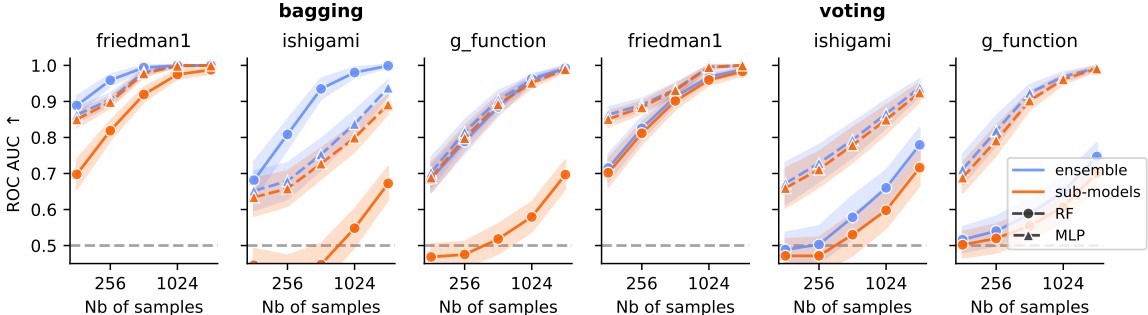

*Figure 9.* **Model-level ensembling improves feature selection performance with *SAGE* .** Area Under the ROC Curve (ROC AUC) for feature selection as a function of sample size ($n$). The ROC curves evaluate the ability to classify features as relevant versus irrelevant, where ground truth relevance is defined by having a non-zero asymptotic importance value (computed at $n = 10^5$). The estimated feature importance score serves as the decision function for the ROC. The plots compare the ensemble strategy (blue) against the sub-models strategy (orange) for RF and MLP architectures across the three benchmark datasets (Friedman 1, Ishigami, G-function). Left panels: Results using bagging (bootstrap training samples). Right panels: Results using voting (random initializations). Higher AUC values indicate that the method assigns higher importance scores to relevant features compared to null features. Results are averaged over 100 random seeds; error bars represent the standard deviation of the ROC AUC.

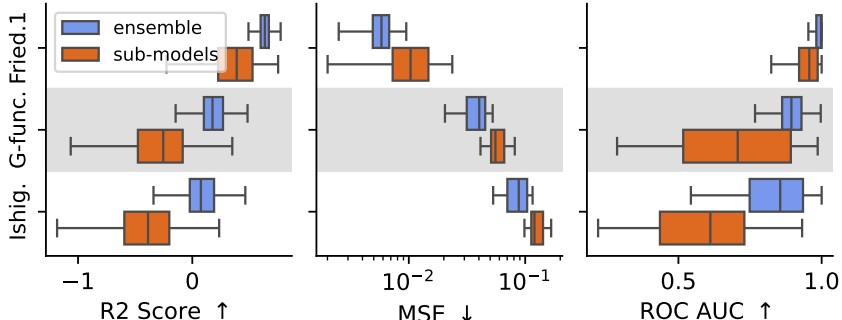

*Figure 10.* **Ensemble-level importance improves estimation across diverse datasets.** Measuring *SAGE* importance directly on the bagging ensemble (blue) consistently outperforms the average of sub-model scores (orange). Higher $R^2$ scores demonstrate the ensemble's superior predictive performance. This improved prediction translates into more accurate numerical estimates for ranking (lower MSE) and more reliable feature selection (higher ROC AUC). These gains are robust across the Friedman 1, G-function, and Ishigami datasets (rows of the plots), which present varying non-linearities and levels of feature interactions. For all datasets, the number of samples was set to n=512. Box plots represent results across 100 random seeds and both MLP and RF architectures.

## C. Analysis of *CFI*

### C.1. Theoretical analysis

Conditional Feature Importance (*CFI* ) quantifies the importance of a feature $j$ by comparing the risk of a model $f$ to its risk under a perturbed input $X^{\pi(j|-j)}$. In this scheme, the $j^{th}$ feature is resampled from the conditional distribution $P(X^j|X^{-j})$ while all other features remain unchanged. For this analysis, we assume an additive noise model $y = f_\star(X) + \epsilon$, where $\epsilon \perp\!\!\!\perp X$ and $\mathbb{E}[\epsilon] = 0$. Under the mean squared error (MSE) loss, the true *CFI* is defined as:

$$\psi^{\text{cfi}}(j) = \mathbb{E}\left[\left(y - f(X^{\pi(j|-j)})\right)^2\right] - \mathbb{E}\left[(y - f(X))^2\right] \tag{12}$$

In practice, the conditional distribution $P(X^j|X^{-j})$ is unknown and must be estimated from data. To characterize the impact of this estimation, we denote $X_n^{\pi(j|-j)}$ as the random variable sampled from the estimated conditional distribution, while $X_\star^{\pi(j|-j)}$ represents the true conditional distribution. The total estimation error for *CFI* is:

$$\psi_n^{\text{cfi}} - \psi_\star^{\text{cfi}} = \frac{1}{n}\sum_{i=1}^n \left(y - f_{\theta,\mathcal{D}_n}(X_n^{\pi(j|-j)})\right)^2 - \frac{1}{n}\sum_{i=1}^n (y - f_{\theta,\mathcal{D}_n}(X))^2$$
$$- \mathbb{E}\left[\left(y - f_\star(X_\star^{\pi(j|-j)})\right)^2\right] + \mathbb{E}\left[(y - f_\star(X))^2\right]$$

Following the same logic as in Appendix A, we can simplify the expression by accounting for the $O_P(n^{-1/2})$ test-set estimation and obtain:

$$\psi_n^{\text{cfi}} - \psi_\star^{\text{cfi}} = \mathbb{E}\left[\left(y - f_{\theta,\mathcal{D}_n}(X_n^{\pi(j|-j)})\right)^2\right] - \mathbb{E}\left[\left(y - f_\star(X_\star^{\pi(j|-j)})\right)^2\right]$$
$$- \mathbb{E}\left[(y - f_{\theta,\mathcal{D}_n}(X))^2\right] + \mathbb{E}\left[(y - f_\star(X))^2\right] + O_P(n^{-1/2}).$$

By applying Proposition 3.4 from Paillard et al. 2025, the error can be decomposed as:

$$\psi_n^{\text{cfi}} - \psi_\star^{\text{cfi}} = \mathbb{E}\left[\left(f_{\theta,\mathcal{D}_n}(X_\star^{\pi(j|-j)}) - f_{\theta,\mathcal{D}_n}(X_n^{\pi(j|-j)})\right)^2\right] + O_P(\mathbb{E}\left[f_{\theta,\mathcal{D}_n} - f_\star\right]) + O_P(n^{-1/2}).$$

This result reveals a fundamental difference in behavior compared to *LOCO* or *SAGE* . For *CFI* , the dependence on model estimation error is captured by the linear term $O_P(f_{\theta,\mathcal{D}_n} - f_\star)$, which typically vanishes faster than the quadratic excess risk terms governing LOCO. Instead, the primary error component stems from the estimation of the conditional distribution. As evidenced by the corollary to Proposition 3.4 in Paillard et al. 2025, for a Lipschitz-continuous model $f_{\theta,\mathcal{D}_n}$, the error in estimating *CFI* can be expressed as:

$$\psi_n^{\text{cfi}} - \psi_\star^{\text{cfi}} = O_P\left(||X_\star^{\pi(j|-j)} - X_n^{\pi(j|-j)}||_2^2\right) + O_P(\mathbb{E}\left[f_{\theta,\mathcal{D}_n} - f_\star\right]) + O_P(n^{-1/2}) \tag{13}$$

This linear dependence implies that ensembling strategies designed to mitigate model excess risk exert a significantly less pronounced impact on *CFI* compared to risk-difference measures such as *LOCO* or *SAGE* . The empirical results presented in Figure 12 and Figure 13 substantiate this theoretical distinction: while the *ensemble* strategy does not underperform relative to *sub-models*, the performance gains are marginal compared to the substantial benefits observed for *LOCO* (Figure 6 and Figure 8) and *SAGE* (Figure 7 and Figure 9). Furthermore, these findings suggest that the error associated with estimating the conditional distribution—denoted as $O_P\left(||X_\star^{\pi(j|-j)} - X_n^{\pi(j|-j)}||_2^2\right)$—is not the limiting factor in practice. This aligns with the "Model-X" assumption (Candes et al., 2018), which posits that modeling the relationships among covariates (i.e., conditional distributions) is generally a more tractable statistical problem than modeling the relationship between input features $X$ and a target $Y$.

## D. Analysis of *PFI*

The analysis of PFI follows that of CFI (Section C.1) with one modification.

PFI replaces the conditional permutation $X^{\pi(j|-j)} \sim P(X^j|X^{-j})$ with a marginal permutation $X^{\pi(j)} \sim P(X^j)$. Since marginal permutation can produce inputs outside the joint distribution's support, loss consistency (Assumption 4.1) does not automatically extend to the permuted inputs. Specifically, when applying Proposition 1 to replace the empirical risk $\mathcal{R}_n(\hat{f}(X^{\pi(j)}))$ with its expectation, bounding the remainder $r_n$ requires $\|\mathcal{L}(\hat{f}(X^{\pi(j)}), Y) - \mathcal{L}(f_\star(X^{\pi(j)}), Y)\|_2 = o_P(1)$, which is only guaranteed under the additional **support positivity** assumption (Assumption 4.8), as for *SAGE*.

Under this additional assumption, the remainder of the proof is identical to *CFI*, yielding:

$$\psi_n^{\text{pfi}} - \psi_\star^{\text{pfi}} = O_P\left(\|X_\star^{\pi(j)} - X_n^{\pi(j)}\|_2^2\right) + O_P(\mathbb{E}[\hat{f} - f_\star]) + O_P(n^{-1/2})$$

The same model is applied to both terms of the risk difference, so the quadratic excess risk cancels.

We replicate the benchmark from Figure 4 using *PFI* instead of *LOCO* , aggregating results over both MLP and RF architectures (Figure 11). Although the ensemble achieves substantially higher $R^2$ scores, these predictive gains barely translate into improved feature importance estimation. This is consistent with the theoretical analysis in subsection 4.4: because PFI has only a linear dependence on model excess risk, reducing the excess risk through ensembling has a much weaker effect on importance estimation error than for risk-difference methods such as *LOCO* or *SAGE* .

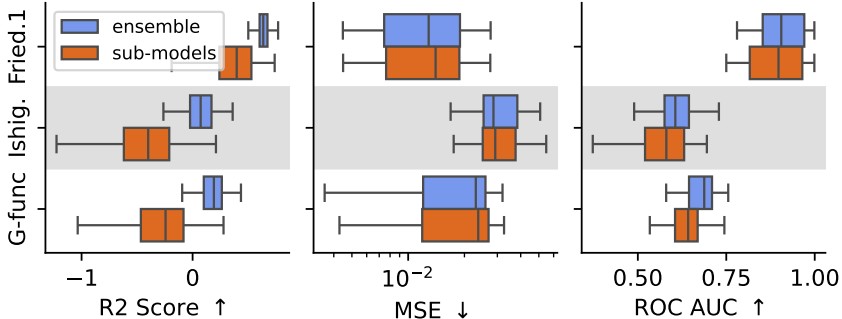

*Figure 11.* **Predictive gains from ensembling do not translate into better *PFI* estimation.** Same experimental setup as Figure 4, aggregating over both MLP and RF architectures. Although the ensemble (blue) achieves substantially higher $R^2$ scores than the sub-models average (orange), the resulting improvements in importance MSE and feature selection ROC AUC are barely visible. This confirms the theoretical analysis in subsection 4.4: because PFI depends only linearly on model estimation error, reduced excess risk does not strongly benefit importance estimation. Results are shown across the Friedman 1, G-function, and Ishigami datasets (rows), with $n = 512$. Box plots represent results across 100 random seeds and both MLP and RF architectures.

## E. Analysis of *IG*

This work focused on methods providing population-level importance scores. Yet, we include an additional analysis on Integrated gradients (*IG*), which is an instance label attribution method. For a differentiable model $f$, the *IG* attributes importance (integrating the gradients along a path) to feature $j$ at input $x$ (relative to a baseline $x'$) following,

$$\text{IG}(x, f) = (x_j - x'_j) \int_0^1 \nabla_j(f(x' + \alpha(x - x')))d\alpha$$

Unlike the non-linear importance methods analyzed in the rest of this work, the analysis for IG concludes immediately because it is a linear operator. For an ensemble $f_{ens} = \frac{1}{B}\sum_{b=1}^B f_b$, computing the IG on the ensemble is mathematically

identical to averaging the IGs of the individual sub-models. By the linearity of differentiation and integration:

$$\mathrm{IG}_j(x, f_{ens}) = (x_j - x'_j) \int_0^1 \nabla_j \left( \frac{1}{B} \sum_{b=1}^{B} f_b(x' + \alpha(x - x')) \right) d\alpha$$

$$= \frac{1}{B} \sum_{b=1}^{B} (x_j - x'_j) \int_0^1 \nabla_j (f_b(x' + \alpha(x - x'))) d\alpha$$

$$= \frac{1}{B} \sum_{b=1}^{B} \mathrm{IG}_j(x, f_b)$$

Thus, for IG, both aggregation strategies yield the exact same attribution. This can also easily be verified empirically.

### E.1. Experiments

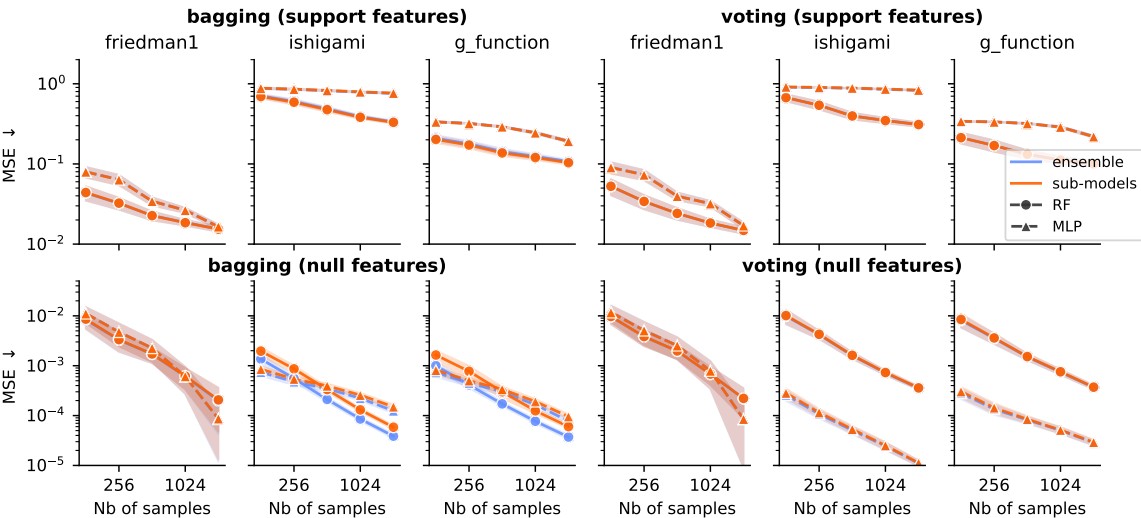

*Figure 12.* **Effect of *ensembling* versus *sub-models* strategy on the estimation error with *CFI* .** Mean Squared Error (MSE) of *CFI* feature importance estimates as a function of sample size ($n$) for Random Forest (RF) and Multi-Layer Perceptron (MLP) architectures. The plots compare two estimation strategies: ensemble (blue), where importance is derived from the aggregated model, and sub-models (orange), where importance scores from individual models are averaged. Top row: MSE for features within the true support. Bottom row: MSE for null features. Columns display results for the three benchmark datasets: Friedman 1, Ishigami, and G-function. The left panels correspond to bagging (ensembling over bootstrap training samples), while the right panels correspond to voting (ensembling over random initializations). Results are averaged over 100 random seeds; error bars represent the standard deviation of the ROC AUC.

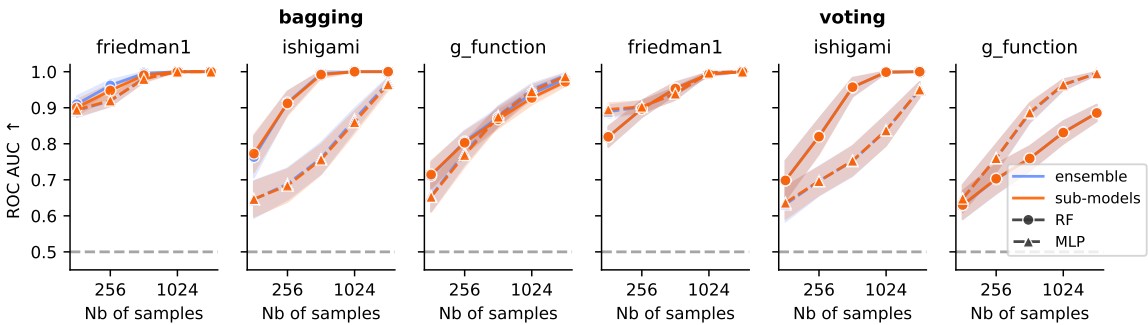

*Figure 13.* **Effect of *ensembling* versus *sub-models* on feature selection performance with *CFI* .** Area Under the ROC Curve (ROC AUC) for feature selection as a function of sample size ($n$). The ROC curves evaluate the ability to classify features as relevant versus irrelevant, where ground truth relevance is defined by having a non-zero asymptotic importance value (computed at $n = 10^5$). The estimated feature importance score serves as the decision function for the ROC. The plots compare the ensemble strategy (blue) against the sub-models strategy (orange) for RF and MLP architectures across the three benchmark datasets (Friedman 1, Ishigami, G-function). Left panels: Results using bagging (bootstrap training samples). Right panels: Results using voting (random initializations). Higher AUC values indicate that the method assigns higher importance scores to relevant features compared to null features. Results are averaged over 100 random seeds; error bars represent the standard deviation of the ROC AUC.

## F. Identification of proteomic signatures for BMI with *SAGE* and *CFI* in the UK Biobank

Complementing the analysis in Figure 5, we used *SAGE* to identify proteomic signatures of body mass index in the UK Biobank. Figure 15 reports the ten features with the largest absolute importance. Given the high computational complexity of *SAGE*, we restricted the evaluation to a random subset of 1,024 test samples per fold. The top-ranked proteins corroborate the findings identified by *LOCO*, notably identifying LEP (Leptin), FABP4 (Fatty Acid Binding Protein 4), and ADM (Adrenomedullin) as key predictors. Furthermore, the distinction between aggregation strategies persists: the *sub-models* approach exhibits larger standard deviations and underestimates the importance of Insulin-like Growth Factor Binding Proteins (IGFBP), particularly IGFBP-1, compared to the *ensemble* strategy.

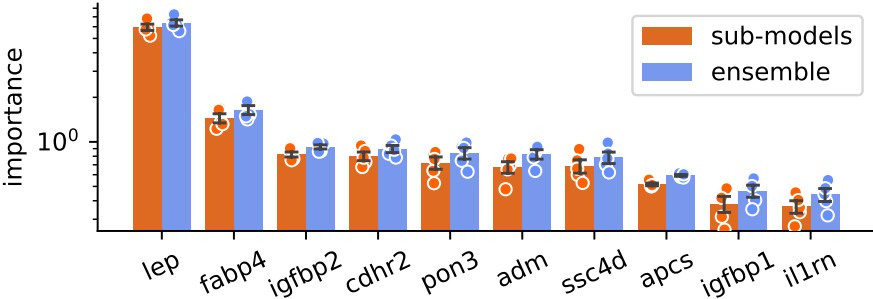

*Figure 14.* **Identification of proteomic signatures for body mass index in the UK Biobank.** Feature importance ranking of the top 10 proteins identified with *SAGE* for the prediction of BMI from 2,922 proteins measured in plasma using the Olink platform ($n = 46,382$ participants). The predictive model is an ensemble comprising 10 *LightGBM* models. Error bars indicate one standard deviation estimated via 5-fold cross-validation. For each fold, the ensemble importance score and the mean importance across individual models are represented by blue and orange circles, respectively.

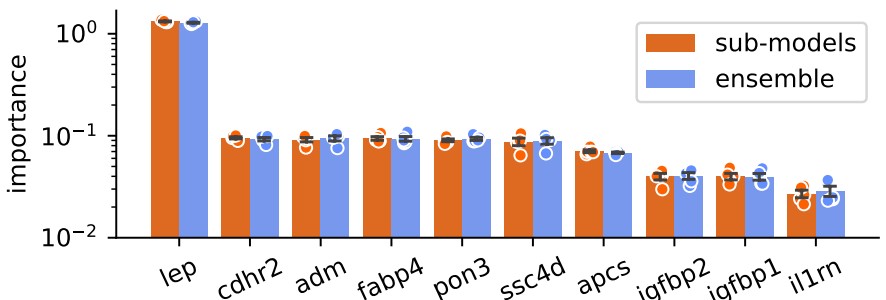

*Figure 15.* **Identification of proteomic signatures for body mass index in the UK Biobank.** Feature importance ranking of the top 10 proteins identified with *CFI* for the prediction of BMI from 2,922 proteins measured in plasma using the Olink platform ($n = 46,382$ participants). The predictive model is an ensemble comprising 10 *LightGBM* models. Error bars indicate one standard deviation estimated via 5-fold cross-validation. For each fold, the ensemble importance score and the mean importance across individual models are represented by blue and orange circles, respectively.

## G. Breast cancer subtype classification (TCGA BRCA)

Following the experimental protocol of Catav et al. (2021) and Janssen et al. (2023), we evaluate the two aggregation strategies on a breast cancer subtype classification task from the TCGA BRCA RNA-Seq gene expression dataset. The dataset comprises 572 patients and 50 genes, where the task is to classify samples into 4 subtypes. Among the 50 genes, 10 have been previously reported in the literature as drivers of breast cancer subtype classification (BCL11A, EZH2, IGF1R, LFNG, BRCA1, SLC22A5, CDK6, BRCA2, TEX14, CCND1) and serve as ground truth; the remaining 40 genes are permuted across patients to break their association with the labels. This setup provides a controlled benchmark with known ground truth for evaluating feature selection performance on real biological data.

We compare two model architectures: a Multi-Layer Perceptron (MLP) and a Logistic Regression with $\ell_2$ regularization. For each architecture, an ensemble of 10 models is constructed. Feature importance is measured using *LOCO* with the

cross-entropy loss. Evaluation is conducted over 10-fold stratified cross-validation, repeated across 10 random seeds.

Figure 16 summarizes the results. Panels (a) and (d) confirm that both architectures achieve good predictive performance (AUC $\approx 0.9$), with the ensemble outperforming sub-models, as expected. Panels (b) and (e) display the fraction of ground-truth driver genes recovered as a function of the number of top-ranked genes selected. The *ensemble* strategy consistently recovers driver genes faster than the *sub-models* strategy, particularly for small values of $K$, indicating that measuring importance on the aggregated model yields a more accurate ranking. Panels (c) and (f) report the AUC for classifying genes as drivers versus non-drivers using each strategy's importance scores as a decision function. The *ensemble* approach achieves higher and more stable AUC scores across seeds. These results on real biological data with established ground truth corroborate the findings from the synthetic experiments: measuring feature importance at the ensemble level leads to improved identification of truly relevant features.

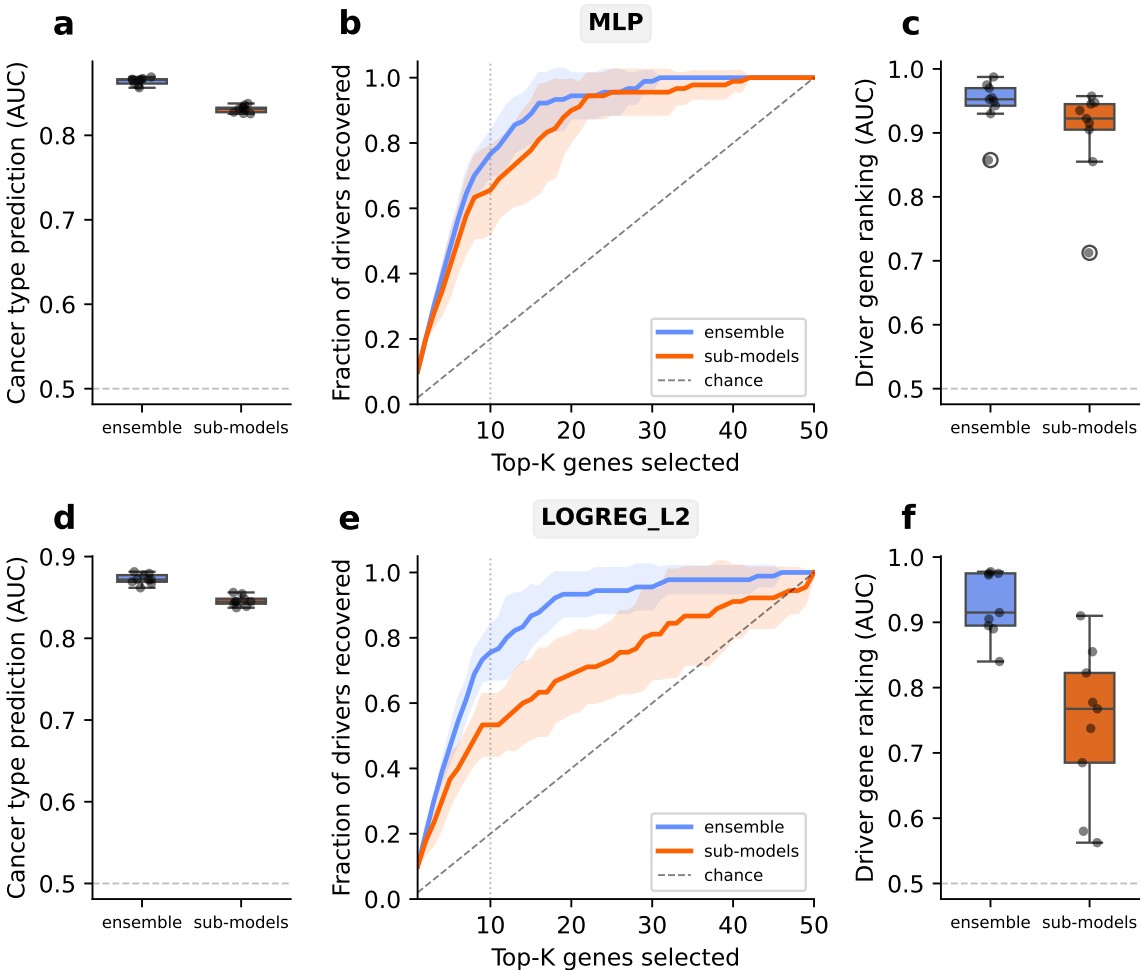

*Figure 16.* **Recovery of breast cancer driver genes on the TCGA BRCA dataset.** Comparison of the *ensemble* (blue) and *sub-models* (orange) strategies for identifying known driver genes using *LOCO* importance. Top row: MLP. Bottom row: Logistic Regression. **(a, d)** Predictive performance (ROC AUC) of the ensemble versus the average performance of sub-models, evaluated on held-out folds. **(b, e)** Fraction of the 10 ground-truth driver genes recovered among the top-$K$ ranked genes. The vertical grey line indicates $K = 10$. Shaded areas represent $\pm 1$ standard deviation across 10 seeds. **(c, f)** AUC for classifying genes as drivers using importance scores as the decision function.

## H. High-dimensional simulation ($d = 100$)

To extend the synthetic benchmarks from section 5 (where $d = 20$) to a higher-dimensional regime, we simulate data with $d = 100$ features and a support of size 25. The target is a non-linear function of the support, built from randomly

sampled combinations of absolute value, sine, cosine, sigmoid, and square transformations (SNR = 1). The model is an overparameterized MLP (2 hidden layers of 256 neurons), with bagging ensembles of 10 models and *LOCO* importance. Ground-truth importance is estimated at $n = 10{,}000$.

As shown in Figure 17, the *ensemble* strategy achieves consistently lower importance MSE and higher feature selection AUC across all sample sizes (results aggregated over 100 random seeds), confirming that the benefits observed in Figure 4 extend to higher dimensions with larger support and overparameterized models.

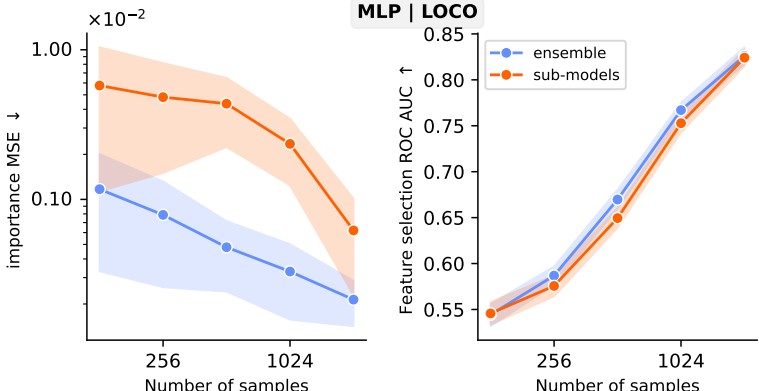

*Figure 17.* **High-dimensional simulation ($d = 100$, 25 support variables).** *Ensemble* (blue) vs. *sub-models* (orange) using *LOCO* importance with an overparameterized MLP and bagging (10 models). **Left:** MSE of importance estimates relative to asymptotic values. **Right:** ROC AUC for feature selection. Averaged over 100 seeds; shaded areas: $\pm 1$ s.d.

## I. Extension to a tabular foundation model (TabICL)

We replicate the benchmark from Figure 4 using TabICL, a state-of-the-art tabular foundation model based on in-context learning. The setup is identical: Friedman 1, Ishigami, and G-function datasets with $d = 20$, $n = 512$, *LOCO* importance, and bagging ensembles of 10 models.

Figure 18 confirms that the *ensemble* strategy yields substantially lower importance MSE. ROC AUC is near-perfect for both strategies, therefore showing little difference.

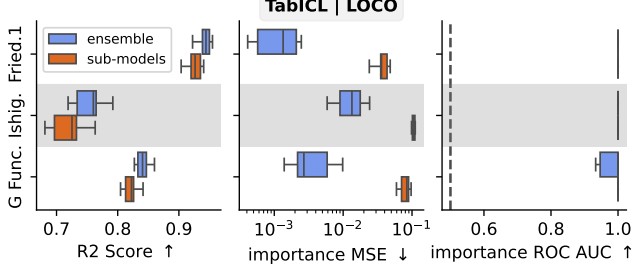

*Figure 18.* **TabICL on the synthetic benchmarks from Figure 4.** *Ensemble* (blue) vs. *sub-models* (orange) using *LOCO* importance with a TabICL foundation model and bagging (10 models), $n = 512$, $d = 20$. **Left:** $R^2$ score. **Center:** Importance MSE (log scale). **Right:** Feature selection ROC AUC.

## J. Stability of importance rankings

Beyond estimation accuracy, we evaluate the stability of importance rankings across cross-validation folds using Spearman's rank correlation, complementing the MSE and ROC AUC metrics from section 5. For each seed, we compute the average pairwise Spearman correlation between importance rankings obtained on different held-out folds.

Figure 19 reports results on the three synthetic benchmarks for both *LOCO* and *SAGE* , using MLP and Random Forest

architectures. The *ensemble* strategy consistently yields higher rank correlations across datasets, sample sizes, and importance methods, indicating more stable importance rankings across data splits.

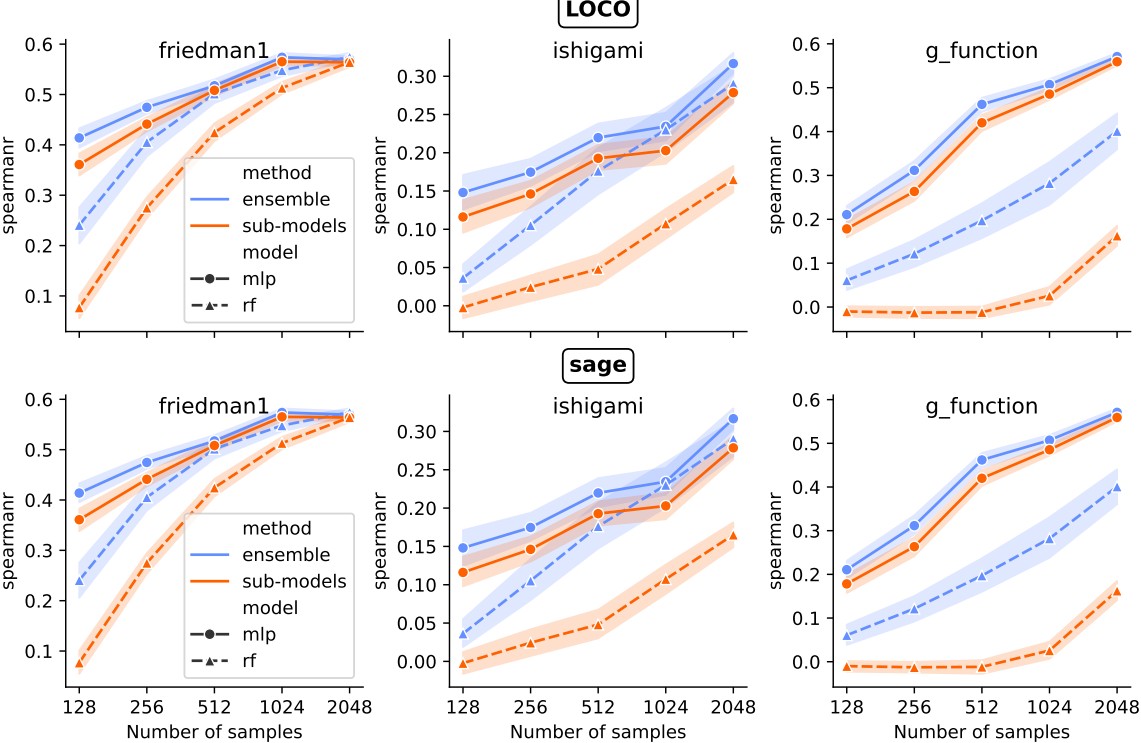

*Figure 19.* **Stability of importance rankings measured by Spearman's rank correlation.** Average pairwise Spearman correlation between importance rankings across cross-validation folds, as a function of sample size. *Ensemble* (blue) vs. *sub-models* (orange) for MLP (solid, circles) and Random Forest (dashed, triangles). Top row: *LOCO* . Bottom row: *SAGE* . Columns correspond to the Friedman 1, Ishigami, and G-function benchmarks ($d = 20$). Averaged over 100 seeds; shaded areas: $\pm 1$ s.d.

