# OpenReview forum: "Aggregate Models, Not Explanations: Improving Feature Importance Estimation"
_ICML.cc/2026/Conference — ICML 2026 regular_

### Official Review · Reviewer_kBfb · 2026-03-04

**Soundness:** 4
**Presentation:** 3
**Significance:** 4
**Originality:** 3
**Overall Recommendation:** 6
**Confidence:** 3

**Summary:**

The paper "Aggregate Models, Not Explanations: Improving Feature Importance Estimation" provides a theoretical analysis and empirical results on the question of whether feature importance values should be derived from a model-level ensemble or from aggregating the importance scores of individual submodels. This is of practical importance in explainable AI due to considerations related to the Rashomon effect. The theoretical analysis focuses on LOCO, SAGE, and CFI. The experiments use Multi-Layer Perceptrons and Random Forests. The datasets are synthetic. The conclusions from the theoretical analysis and the experiments align in showing that ensembling at the model level improves upon aggregating individual models’ importances. The text is biologically motivated and provides a study on the identification of proteomic signatures for BMI in the UK Biobank.

**Compliance With Llm Reviewing Policy:**

Affirmed.

**Final Justification:**

Dear all, after the rebuttal period I changed the score to "6: Strong Accept". This paper is technically sound and provides both theoretical analysis and empirical experiments to support its claims. The results are useful for the XAI community, and the Rebuttal by Authors added new informative experiments and results.

**Key Questions For Authors:**

1. What should be expected if similar analyses and experiments were performed for permutation importance?

2. Some popular feature importance methods are not model-agnostic, and consequently, for an ensemble, the only option would be to aggregate the importance values of the submodels. In such cases, considering your results, would it be preferable not to use model-specific methods?

3. Have you considered the definitions of stability and reliability used in the feature selection literature, including stability by index/subset, stability by rank, and stability by weight, as well as metrics such as the Jaccard index, Sørensen-Dice coefficient, percentage overlap, Kuncheva index, and Pearson correlation coefficient, in addition to the use of visualization techniques for feature importance analysis, including t-SNE and UMAP?

**Limitations:**

The main limitations of the paper are adequately described in the "Limitations and future work" section: the limited scope, which excludes heuristic methods, and the restricted empirical validation to tabular data. The societal impacts are positive, as this paper improves our understanding of popular interpretability algorithms.

**Strengths And Weaknesses:**

Soundness: This paper is technically sound and provides both theoretical analysis and empirical experiments to support its claims, which ultimately align. The authors used only synthetic data for their main experiments, which may seem like a major limitation. However, this is justified by the need to know beforehand the true feature importance values in order to validate the results. The authors also provided an additional experiment using real-world data on proteomic signatures for BMI in the UK Biobank. The paper’s main research question is interesting and relevant. Regarding related work, the paper claims that there are no definitions of stability, which is not adequately discussed, as there are references on stability and reliability in the context of feature selection. The main limitation of the work is its limited scope, excluding heuristic methods and restricting empirical validation to tabular data, thus not addressing some of the most popular methods used in practice or all use cases. However, the paper addresses and justifies these choices appropriately.

Presentation: The paper is complete, well-organized, and properly written. The text regarding the theoretical analysis could be made more approachable by including more introduction to the key components and terms, and by providing intuition for the results. The text claims more than once that there are no definitions of stability, but there is extensive literature on stability and reliability in the context of feature selection that would be applicable in this context.

Significance: The authors tackled a narrow research question, but one with theoretical and practical significance for the understanding and deployment of feature importance methods. They also provide a clear guideline at the end (ensembling at the model level over aggregating individual model importance scores). The contribution is limited by the scope of the work, which only used LOCO, SAGE, and CFI. Expanding the work to other mainstream feature importance methods would increase the significance of this study.

Originality: Despite being a prevalent issue in the literature and in practice, I have not seen many studies tackling the central question of this paper so directly. To the best of my knowledge, the analyses, experiments, and results presented by the authors are new.

---

> ### Author Rebuttal · Authors · 2026-03-29
>
> We thank the reviewer for their valuable feedback. We address the questions regarding PFI, model-agnostic methods, and stability below.
>
> ## 1. Extension to Permutation Feature Importance (PFI) (Q1)
> For PFI, the analysis is very similar to CFI but requires an additional assumption, support positivity (4.8) similarly to SAGE.
>
> PFI is defined as a risk difference, $\psi_{\text{pfi}}^\star(j) = R(f^\star(X^{\pi(j)})) - R(f^\star(X))$, where $X^{\pi(j)}$ replaces $X_j$ with an independent draw from the marginal $P(X_j)$ (obtained by permutations). The proof sketch follows the same structure as our CFI analysis (Section 4.3):
>
> 1. **Same model, perturbed inputs.** As with CFI, the same model $f^\star$ is applied to both $X^{\pi(j)}$ and $X$. As a result, the excess risk terms $\mathcal{E}(f_\theta)$ that appear on both sides of the difference computed by PFI are the same. Consequently, they cancel out, leaving a linear dependence $O_P(\mathbb{E}[f_\theta - f^\star])$, like CFI.
>
> 2. **Where PFI differs from CFI.** For CFI, the perturbation is obtained by  sampling $X_j$ from the conditional $P(X_j | X_{-j})$, which preserves the data distribution's. On the contrary, PFI samples from the marginal $P(X_j)$, which can produce inputs $X^{\pi(j)}$ that lie outside the support of the joint distribution (e.g., combining a marginally sampled $X_j$ with $X_{-j}$ values that would never co-occur). As a reult, the *loss consistency* assumption (4.1), which is needed for the control of the remainder term by ensuring that $f_\theta(X)$ is a risk minimizer does not guarantee that  $f_\theta(X^{\pi(j)})$, wehereas this held for CFI. Therefore, similar to SAGE, an additional assumption on *support positivity* (4.8) is needed to ensure that $X^{\pi(j)}$ lies in the support and that the loss consistency applies. A discussion on this assumption is provided in the submission L279.
>
> 3. **Conclusion.** Under this additional assumption, PFI behaves like CFI: the linear dependence on the excess risk means Jensen's inequality provides no improvement for model-level ensembling. We expect the two strategies to perform comparably, as confirmed for CFI in Figure 11 and see link below for PFI.
>
> We will include this analysis as part of the section on CFI in the revised manuscript.
> This theoretical analysis is also empirically validated as shown in this figure, which reproduces the figure 4 from the submission, but here with PFI: https://imgur.com/a/zzL6i6P
> Contrary to LOCO and SAGE, there is hardly any difference in the importance metrics (MSE and AUC) between the two strategies for PFI.
>
> ## 2. Model-specific methods
>
> While model-level ensembling can be addressed for some model-specific methods (e.g., it is mathematically trivial for linear operators like Integrated Gradients, as discussed with Reviewer Hh42), our focus on model-agnostic methods is linked to our theoretical findings.
>
> Our analysis shows that the error in feature importance estimation is driven by the predictive model's excess risk. Consequently, obtaining the most accurate feature importance requires using the most accurate predictive model possible for a given dataset. Model-agnostic methods allow practitioners to adaptively select the best-performing architecture rather than being restricted to a specific model class just to access its built-in importance metric. We will clarify this motivation in the introduction.
>
> ## 3. Stability and reliability
> To clarify our statement that stability is "rarely formally defined," this was a direct reference to the motivation in Donnelly et al. (2023). In their work, they note the lack of consensus around a single definition in the literature and explicitly decide not to adopt one, leaving the concept open.
>
> Because stability can refer to variance, rank correlation, or selection frequency, it must be explicitly defined before any formal analysis. Ultimately, the goal is to correctly estimate feature importance, meaning we must account for both bias and variance rather than variance alone. We chose to focus on Mean Squared Error (MSE) because it captures both. Furthermore, if importance values are estimated accurately (low MSE), the resulting feature ranking will inherently be correct and stable. Thus, MSE is linked to standard selection metrics while addressing the richer problem of quantifying the importance values.
>
> We value the suggestion to evaluate ranking stability; we computed the **Spearman rank correlation**. We chose this metric as it allows us to compare selection consistency across different thresholds. We computed this correlation directly from the results (importances) of the experiments presented in the submission. The correlation is measured between all pairs of folds, for all three datasets, with error bars representing the variation across 100 random seeds (see attached figure: https://imgur.com/a/r56psfi
> ). The results show that with this metric as well, aggregating models predictions is preferrable to importance scores.

---

> > ### Author Rebuttal · Reviewer_kBfb · 2026-04-01
> >
> > Thank you for the rebuttal. The new experiments and results are very informative.

---

### Official Review · Reviewer_ktp6 · 2026-03-11

**Soundness:** 2
**Presentation:** 3
**Significance:** 3
**Originality:** 3
**Overall Recommendation:** 5
**Confidence:** 3

**Summary:**

Many feature importance methods use the expected loss of a predictive model in order to compute scores for how much a given variable affects a response variable (usually by taking the difference of two models, where only one model is trained using data from the feature of interest). This work considers two ways of aggregating multiple predictive models to obtain more accurate scores. The first, *ensemble*, is to aggregate the model outputs themselves, and then to compute expected loss with respect to the ensemble model in order to determine feature importance. The second, *sub-models*, is to compute the expected loss from each individual model and then to average associated scores. Because of nonlinear losses, the two are distinct. This work provides a theoretical analysis of the error bounds to show that the first approach (making an ensemble model) is often beneficial for three specific feature importance methods: ablation importance, aka leave-one-covariate-out (LOCO), Shapley additive global importance (SAGE), and conditional feature importance. Experiments on synthetic data validate this finding and an additional real world experiment is run, such that importance scores are reported for blood proteins towards BMI as a response variable.

**Compliance With Llm Reviewing Policy:**

Affirmed.

**Final Justification:**

Author rebuttal handled my main concerns.

**Key Questions For Authors:**

Q1. In the first bullet point of Remark 4.5, it is stated that aggregating over the importance scores of different models may reduce the stochastic remainder term but doing so does not affect the excess risk term, $\mathcal{E}$. However, doesn't aggregating submodel scores also mitigate the component $\mathcal{E}_{opt}$, as stated in the introduction? The explanation that is offered is not clear to me. Otherwise, if aggregating scores does reduce $\mathcal{E}\_{opt}$, then I think that this term needs to be handled more carefully in the analysis in order to determine the conditions for which ensemble is better than submodels.

Q2. Do the authors have additional insights about how to control the asymptotic errors from the components of the excess risk $\mathcal{E}$? Maybe answering this w/r/t $\mathcal{E}\_{opt}$ is equivalent to clarifying my confusion in Q1.

Q3. If all the above can be appropriately handled, it seems that the ensemble vs. submodels asymptotic theory (ex. Prop 4.7 and Thm 4.10)  can be generalized for many more feature importance methods. Have the authors thought about the possibility of refactoring conditions to prove theory for more general families of methods, rather than having a specific result for each method?

Q4. Why do the authors think that CFI does not exhibit much difference in results between the two strategies on the real world dataset?

**Limitations:**

yes

**Strengths And Weaknesses:**

I evaluate strengths and weaknesses across a few components, listing suggestions where applicable. I have some clarification questions about the theory, which I'd like to hear back from the authors (most importantly, Q1 in the questions section). Assuming this is well handled, the paper's principal limitation is its evaluation on real data, since the considered dataset lacks sufficient ground truth compared to similar benchmarking evaluations in the literature.

### Significance and Originality
The paper tackles an interesting problem in the feature importance literature and it offers both theoretical and empirical insights into effective aggregation, which both conclude that ensembling at the model-level is typically advantageous, compared to aggregating scores of submodels. To the best of my knowledge, the asymptotic analysis comparing the two aggregation types has not been previously studied, although I have not checked all of the related works. The theoretical contributions use previous works' characterization of excess risk, but the authors derive novel insights, and have applied the analysis to a few commonly used feature importance methods. The biggest weakness for the theoretical analysis is that it does not offer asymptotic insights into the individual components of the excess risk, instead treating them jointly as one term. Overall, my initial score is a weak accept because I have some questions about the theory (see next section), and I think that the experiment on real data is not quite to the standard of similar papers at ML conferences. I discuss two points for this experiment below, and how I think the evaluation can be strengthened.

**Dataset choice**: The simulated experiments are convincing, and I think that the current BMI real world experiment offers some evidence that the *ensemble* strategy outperforms *sub-models*, but is not completely convincing, largely due to the fact that the scientific knowledge that we have about the ground truth proteins is limited. As such, the current results somewhat read like it is reporting two sets of results (one for each aggregation strategy). In particular, the main argument for the superiority of the ensemble approach is the fact that LOCO (and SAGE in the supplement) attribute higher importance to insulin growth proteins (IGFBP-1 and -2) when using ensemble. This seems desirable, but a relatively limited insight, compared to the standard benchmarking experiments in ML papers for assessing feature importance, such as the BRCA dataset used in the SAGE paper (Covert et al. 2020) and other methods tailored for explaining the data: MCI (Catav et al. 2021) and UMFI (Janssen et al. 2023). Ground truth genes that are biologically associated to breast cancer, e.g. BRCA, have been extensively studied so that 10 of the genes in the dataset can be considered as ground truth features. For even better interpretability, one can also permute the 40 other genes across patients in the datasets, to ensure that these are unassociated to the response (Janssen et al. 2023). Demonstrating the effectiveness of ensemble over submodels on the BRCA dataset (or on another dataset with an established ground truth) would be more compelling for real world practicality.

**Method choice:** The paper considers three popular feature importance methods for both theoretical and empirical consideration. One point to note is that Shapley-based and ablation methods are known to suffer from correlation distortion (Verdinelli et al. 2024), since feature importance of highly correlated featured tend to $0$, even if the variables are causally informative towards the response. As such, they may not be well suited for "explaining the data" when there are many highly correlated variables, despite the fact that the methods are commonly used for this in the literature. It could therefore be interesting to try these methods, at least for the real world experiments that purport to learn the relevant variables, to see if ensembling further improves performance. One would expect that the best performance is achieved by the ensemble version of the "true-to-the-data" methods. Indeed, note that SAGE, ablation, and conditional permutation importance (which I believe precisely represent the methods considered in this paper) are outperformed by MCI and UMFI across experiments, including on BRCA.

### Soundness
I have one concern about the theoretical analysis in section 4. See Q1

### Presentation
The paper is well written and presented, but I have a few suggestions that I think would help readability:
- Section 2 (Problem Setting) and Section 4 (Theoretical analysis) are very closely connected, as equation (2) is central to the error analysis, and the two different types of aggregation are introduced. Consider putting the related work section (currently Section 3) before Problem setting or towards the end of the paper to have these two sections. Also consider emphasizing the two types of aggregation as definitions for visibility. I found myself scrolling up and down the paper a few times to familiarize myself with the terms.
- It wouldn't hurt to define $\mathcal{L}$ as the model's loss in the problem setting
- Similarly, I may have missed it, but I don't believe that the $\mathcal{E}(f_b)$ term had been explicitly defined before its use in Remark 4.5, which seems to connote the excess risk of a specific submodel b.
- It would be more interpretable to write the results in Prop 4.7 and Thm 4.10 to have absolute values over the difference of estimated feature importance score and "true" score

### Refs

Covert, Ian, Scott M. Lundberg, and Su-In Lee. "Understanding global feature contributions with additive importance measures." Advances in neural information processing systems 33 (2020): 17212-17223.

Catav, Amnon, et al. "Marginal contribution feature importance-an axiomatic approach for explaining data." International Conference on Machine Learning. PMLR, 2021.

Janssen, Joseph, Vincent Guan, and Elina Robeva. "Ultra-marginal feature importance: Learning from data with causal guarantees." International conference on artificial intelligence and statistics. PMLR, 2023.

Verdinelli, Isabella, and Larry Wasserman. "Feature importance: A closer look at shapley values and loco." Statistical Science 39.4 (2024): 623-636.

---

> ### Author Rebuttal · Authors · 2026-03-29
>
> We thank the reviewer for the constructive feedback. We address the empirical evaluation and theoretical clarifications below.
>
> ## 1. TGCA BRCA experiment
> We agree that validating our findings on a dataset with established ground truth strengthens the empirical claims. Following your suggestion and similar benchmarking evaluations [1, 2], we conducted a new experiment using the breast cancer (BRCA) subtype classification task from RNA-Seq gene expression profiles (data available via the repository in [1]). See figure: https://imgur.com/a/Mxrt4rT
> The problem is a 4-way classification task using 50 input genes. Based on domain knowledge documented in prior literature, 10 of these genes are established biological drivers (ground truth). Following the exact procedure in [1], we permuted the remaining 40 genes to break their dependency with the labels. We then benchmarked how well each strategy recovers the 10 ground-truth genes:
>  - a. & d. We start by demonstrating the ability of the considered models to classify breast cancer subtypes. Both logistic regression and MLP achieved good predictive performance, with AUCs close to 0.9 .
>  - b. & e. We then rank the genes by LOCO importance for each strategy (ensemble, sub-models) and plot the fraction of ground truth genes recovered in the top K as a function of K, with K ranging from 1 to 50 (total number of genes).
>  - c. & f. Finally, we report the AUC scores for classifying driver genes (ground truth) using each method's importance scores.
>
>
> ## 2. Clarification on $\mathcal{E}_{opt}$ and Excess Risk (Q1 & Q2)
> The optimization error $\mathcal{E}_{opt}$ is a component of the model's excess risk $\mathcal{E}(f)$, which our analysis identifies as the driving term in the importance estimation error.
>
> The difference between the two strategies relies on Jensen's inequality for convex loss functions:
>  - **Importance-level aggregation (sub-models)**: Averaging importance scores effectively averages the individual excess risks: $\frac{1}{B} \sum_{b=1}^B \mathcal{E}(f_b)$. Because this is a simple linear average, it reduces variance but leaves the optimization error unchanged.
>  - **Model-level aggregation (ensemble)**: Ensembling predictions evaluates the excess risk of the ensemble itself: $\mathcal{E}(\frac{1}{B} \sum_{b=1}^B f_b)$. Because standard loss functions are strictly convex, Jensen's inequality guarantees that $\mathcal{E}(\frac{1}{B} \sum_{b=1}^B f_b) < \frac{1}{B} \sum_{b=1}^B \mathcal{E}(f_b)$.
>
> Therefore, only model-level ensembling effectively reduces $\mathcal{E}_{opt}$ and the thereby the leading bias term.
>
> Regarding Q2, our analysis is non-asymptotic and shows that any technique that reduces the excess risk (improves the predictive power) of the estimator (such as better hyperparameter tuning or faster convergence rates) will improve the importance estimation.
>
> ## 3. CFI behavior and generalization (Q3 & Q4)
> The results presented in the submission suggest that there is no general theorem that covers all feature importance methods, although some share similarities (Q3). Because importance methods have distinct mathematical formulations, they require different analytical approaches, rely on different assumptions, and lead to different conclusions regarding ensembling.
>
> The contrast between CFI (and PFI in response to reviewer kBfb) and methods like LOCO or SAGE (Q4) illustrates this well. For instance, analyzing SAGE requires specific assumptions extending model convergence to perturbed inputs. More importantly, LOCO and SAGE rely on quadratic risk differences between **different models**. This leads to the estimation error depending on the models' excess risk and explains why model-level ensembling strictly reduces bias via Jensen's inequality.
>
> By contrast, CFI evaluates the **same model** $f_*$ on both original and permuted inputs: $\psi_*^{cfi}(j) = \mathcal{R}(f_*(X^{\pi(j|-j)})) - \mathcal{R}(f_*(X))$. Because the same model is evaluated in both terms, the quadratic excess risk terms cancel out during the error analysis. This leaves a purely linear dependence on the model's error, $O_P(f_* - f^\star)$, as shown in Eq. 13.
>
> Because this error term is linear, both importance-level and model-level aggregation affect it similarly. Without the quandratic excess risk term present in the error for LOCO and SAGE, model-level ensembling does not provide a strict bias reduction for CFI. This explains why the empirical benefits of model-level ensembling are much less pronounced for CFI (figure 11), and highlights why analyzing different importance methods requires specific theoretical treatments. We will clarify this in the revision.
>
>
> ## References
>  1. Janssen J. et al. Ultra-marginal Feature Importance: Learning from Data with Causal Guarantees, AISTATS, 2023
>  2. Catav A. et al. Marginal Contribution Feature Importance - an Axiomatic Approach for Explaining Data ICML 2021

---

> > ### Author Rebuttal · Reviewer_ktp6 · 2026-04-01
> >
> > 1. Thank you for the additional BRCA experiment with LOCO, it's quite interesting and a good result for showing ensemble outperforms sub-models.
> > 2, 3. Thanks for clarifying, I see now.
> >
> > I've raised my score accordingly.

---

### Official Review · Reviewer_Hh42 · 2026-03-11

**Soundness:** 3
**Presentation:** 3
**Significance:** 2
**Originality:** 2
**Overall Recommendation:** 4
**Confidence:** 4

**Summary:**

This paper addresses feature importance instability in complex machine learning models caused by data sampling and algorithmic stochasticity. The authors investigate whether to aggregate importance scores from individual sub-models or derive importance from an ensemble model. Their theoretical analysis reveals that for expressive models, excess risk dominates estimation error. Consequently, aggregating at the model level reduces bias more effectively than averaging individual explanations. Extensive experiments on benchmarks like Friedman 1 and UK Biobank proteomic data validate that model-level ensembling improves accuracy in feature importance estimation and selection performance compared to traditional sub-model aggregation methods.

**Compliance With Llm Reviewing Policy:**

Affirmed.

**Final Justification:**

I thank the authors for the added experiment and explanations.

**Key Questions For Authors:**

N/A

**Strengths And Weaknesses:**

I am not very sure how I should evaluate this paper. I believe that the authors have done their study “correctly,” and the conclusion—that one should aggregate models instead of explanations when possible—is sound.

My concern is that this result is trivial.

In Shapley value-based explanation methods, such as SAGE, the core to such methods is the valuation function, v, which returns the importance of a subset of the input. In practice, this function is approximated differently depending on the implementation. However, it is very clear that the “correctness” of a feature attribution explanation depends on the quality of the valuation function. As the valuation function can never outperform the prediction function (as it only uses a “partial” input, whereas the prediction function uses the entire input), there is inherent uncertainty in the valuation function.

Thus, when the model, or the prediction function, is weak, the valuation function will be even weaker. This yields high variability in the explanations. Aggregating such weak results still yields weak outcomes. On the other hand, when an ensemble is created, the prediction function becomes stronger, and so do the valuation functions. Thus, we have more stable results. The analysis for LOCO is similar.

The problem gets interesting when the explanation method is not Shapley value-based or dependent on evaluating a subset of inputs, such as with LOCO. Specifically, if the explanation method is Integrated Gradients, then the question becomes harder to answer. Sure, one first needs to come up with a way to aggregate a few neural networks into a single network so IG can be applied to the “ensemble” as well. But this is likely doable, by, e.g., adding a small MLP to collect predictions from individual models. But in such a setting, whether aggregating IG explanations from individual small networks is worse than directly computing IG on the aggregated network remains uncertain.

---

> ### Author Rebuttal · Authors · 2026-03-29
>
> We thank the reviewer for their assessment. We address the concern about the significance of the contribution and the question on Integrated Gradients below.
>
> ## 1. On the significance of the contribution
> ### An actively studied problem but explored in the opposite direction.
> The question of how to estimate feature importance in the presence of multiple models has received considerable attention in recent literature. To focus on two highly cited works, [Fisher et al., 2019] proposed studying importance across an entire class of prediction models simultaneously, and [Donnelly et al., 2023] introduced the Rashomon Importance Distribution, aggregating importance scores from multiple models to address instability. Contrary to the reviewer's intuition that aggregating "weak" explanations "still yields weak outcomes," both works demonstrated that importance-level aggregation (averaging explanations from individual, models) does improve over single-model importance. This established it as the standard approach in the literature. However, neither considered the alternative of aggregating predictions before measuring importance. Our formal analysis reveals that this **overlooked alternative is more effective** and identifies the excess risk as the driving mechanism.
>
> ### A complex problem that requires formal analysis
> Without formal analysis, one could reasonably argue (as the reviewer does) that a better estimator yields better explanations, or conversely (as prior works suggest) that reducing variance by averaging explanations from multiple models is the right approach. However, going beyond these intuitions **requires a formal analysis** which evidences the driving error terms and why acting on the excess risk is the most efficient way to address this problem, especially when considering complex models.
>
> ### Conclusions are method dependent
> As additional evidence that these findings are not trivial, our analysis shows that the conclusions depend on the method. Indeed, it reveals that, contrary to LOCO and SAGE, the benefits of aggregating models are much **less pronounced for CFI** (section 4.3) because of the linear dependence on the model's error, rather than quadratically for LOCO and SAGE. This is the case although CFI is "dependent on evaluating a subset of inputs".
>
>
> ## 2. Integrated Gradients (IG)
> We appreciate the suggestion to discuss IG. Unlike the non-linear importance methods analyzed in our submission, **the analysis for IG concludes immediately** because it is a linear operator.
>
> For an ensemble $f_{ens} = \frac{1}{B} \sum_{b=1}^B f_b$, computing the IG on the ensemble is mathematically identical to averaging the IGs of the individual sub-models. For a differentiable model $f$, IG attributes importance (integrating the gradients along a path) to feature $j$ at input $x$ (relative to a baseline $x'$). By the linearity of differentiation and integration:
>
> $$\text{IG}\_j(x, f\_{ens}) = (x\_j - x\'_j) \int_0^1 \nabla\_j \left( \frac{1}{B} \sum\_{b=1}^B f\_b(x' + \alpha(x - x')) \right) d\alpha$$
> $$\quad\quad=  \frac{1}{B} \sum\_{b=1}^B (x\_j - x'\_j) \int\_0^1 \nabla\_j (  f\_b(x' + \alpha(x - x'))) d\alpha$$
> $$\quad\quad= \frac{1}{B} \sum\_{b=1}^B \text{IG}\_j(x, f_b)$$
> Thus, for IG, both aggregation strategies yield the exact same attribution. We will include this analysis in the discussion.
>
> This can easily be verified with a small code snippet
> ```
> import torch
> import torch.nn as nn
> from sklearn.datasets import make_regression
> from captum.attr import IntegratedGradients
>
> class EnsembleMLP(nn.Module):
>     def __init__(self, models):
>         super().__init__()
>         self.models = nn.ModuleList(models)
>
>     def forward(self, x):
>         outputs = [model(x) for model in self.models]
>         return torch.stack(outputs).mean(dim=0)
>
> # 1. Create two randomly initialized MLPs
> torch.manual_seed(0)
> model_1 = nn.Sequential(
>             nn.Linear(5, 16),
>             nn.ReLU(),
>             nn.Linear(16, 1)
>         )
> torch.manual_seed(1)
> model_2 = nn.Sequential(
>             nn.Linear(5, 64),
>             nn.ReLU(),
>             nn.Linear(64, 1)
>         )
>
> ensemble = EnsembleMLP([model_1, model_2])
>
> # 3. Generate data
> X_np, y_np = make_regression(n_samples=100, n_features=5, random_state=0)
> X = torch.tensor(X_np, dtype=torch.float32)
>
> x_input = X[0:1].requires_grad_()
> baseline = torch.zeros_like(x_input)
>
> # 4. Compute IGs for ensemble and sub-models
> ig_1 = IntegratedGradients(model_1).attribute(x_input, baseline)
> ig_2 = IntegratedGradients(model_2).attribute(x_input, baseline)
>
> sub_models_result = (ig_1 + ig_2) / 2
> ensemble_result = IntegratedGradients(ensemble).attribute(x_input, baseline)
>
> max_diff = torch.max(torch.abs(sub_models_result - ensemble_result)).item()
> print(f"\nMax abs diff: {max_diff:.6f}")
>
> >>> Max abs diff: 0.000000
> ```
> ## References
> 1. Fisher et al., All models are wrong, but many are useful. JMLR, 2019.
> 2. Donnelly et al. The Rashomon Importance Distribution. NeurIPS, 2023.

---

> > ### Author Rebuttal · Reviewer_Hh42 · 2026-04-01
> >
> > Thanks. The IG experiment is informative. I have adjusted my score.

---

### Official Review · Reviewer_FnQq · 2026-03-12

**Soundness:** 3
**Presentation:** 3
**Significance:** 2
**Originality:** 2
**Overall Recommendation:** 4
**Confidence:** 3

**Summary:**

The paper shows that model-level ensembling can provide more accurate variable importance estimates by reducing the leading error term. The authors theoretically and empirically analyze two strategies for aggregating explanations: computing importance from a single ensemble predictor (ensemble) and averaging importance scores derived from individual models (sub-models). The analysis is conducted for several importance methods, including LOCO, SAGE, and CFI. And are supported by experiments on the Friedman1, G-function, and Ishigami synthetic datasets, as well as a real-world dataset from the UK Biobank.

**Compliance With Llm Reviewing Policy:**

Affirmed.

**Final Justification:**

The additional experiments on higher-dimensional settings and transformer-based models, along with the clarification on the novelty of the proposed approach during the rebuttal, have addressed my main concerns.

**Key Questions For Authors:**

1. In the UK Biobank experiment, the authors first select the top 50 proteins using univariate associations before computing feature importance. Since feature importance is intended to identify important variables, could this pre-selection bias the analysis or potentially exclude relevant features?
2. The experiments mainly focus on tabular datasets and relatively simple models (e.g., MLPs and tree-based models). How well do the conclusions generalize to modern deep learning models, such as transformers or vision models, or to higher-dimensional settings?

**Limitations:**

Yes

**Strengths And Weaknesses:**

Strengths And Weaknesses
Strengths
1. The paper is clearly written and the main finding is supported by both theoretical analysis and empirical experiments, making the paper easy to follow.
2.The work may provide actionable guidance for practitioners. Since explanations can change when the underlying model changes, understanding how to obtain more stable and reliable feature importance estimates is an important problem for the community.
3. The paper evaluates multiple feature importance methods. The analysis covers several recent methods, including SAGE (2020), LOCO, and CFI (2023), demonstrating that the proposed insights apply across different methods.

Weaknesses
1. The experimental evaluation is quite limited. The experiments mainly use relatively simple models such as MLPs and tree-based models and focus on tabular datasets with a small number of features (d = 20).
2. Since ensemble models generally improve predictive performance, it might be expected that computing feature importance on ensemble predictors would also improve the accuracy of importance estimates. Although the paper provides theoretical justification for this, the main insight may have limited novelty.
3. There is a notion mismatch between equation 2 and A.1.(1) equation regarding the them $r_n$. Also in theorem 4.10, I think it should be [d] instead of D since it is the feature index set.
4. For the experiment on the UK Biobank dataset, it would be helpful to further evaluate whether the identified top 10 proteins align with biological knowledge or domain expertise, or to analyze how the results change when different subsets of proteins are removed.

---

> ### Author Rebuttal · Authors · 2026-03-29
>
> We thank the reviewer for the constructive feedback and address the questions below.
> ## 1. Higher dimensions, transformers, and vision models
> We conducted two new experiments to address the reviewer's concern about limited experimental scope.
>
> **d=100 non-linear simulation**: We designed a non-linear regression task with d=100 features (25 active). The target is a randomly sampled combination of non-linear transformations (sin, cos, square, abs, tanh) applied to active features, with correlated Gaussian inputs ($\rho=0.5$). Using an overparameterized MLP with bagging (10 members) over 100 seeds, the ensemble strategy consistently achieves lower importance MSE (w.r.t. a ground truth estimated with $n=10^5$ ) and higher AUC for support recovery across all sample sizes (128–2048). see figure: https://imgur.com/a/UrrwIjl
>
> **TabICL foundation model (Transformers)**: We ran the Figure 4 benchmark using TabICL [1], a state-of-the-art pre-trained transformer. Using bagging (10 members) on Friedman1, Ishigami, and G-function (d=20), the ensemble strategy achieves substantially lower importance MSE, with an order-of-magnitude improvement. ROC AUC is near-perfect for both strategies. see figure https://imgur.com/a/TO36Gmu
>
> **Vision models**: We focus on tabular data because its features have a consistent semantic meaning across subjects, unlike raw image pixels. This consistency is required to define population-level importance, allowing us to obtain statistical guarantees (LOCO, CFI ...), which are often overlooked in modern attribution. To apply such framework to vision, one must first extract representations or concepts (e.g., DINO embeddings or Concept Bottleneck Models) that hold consistent meaning across samples, effectively reducing the task back to a tabular setting.
>
> ## 2. Clarification on novelty
> We would like to clarify that our contribution is not simply that ensemble models yield better importance than a single model. Rather, we compare two distinct aggregation strategies:
> 1. **Sub-models**: compute importance on each model separately, then average the scores.
> 2. **Ensemble**: aggregate predictions first, then compute importance once on the ensemble.
>
> The sub-models strategy is the current standard advocated by influential recent work: [Fisher et al. 2019] proposed studying importance across a class of models, and [Donnelly et al. 2023] introduced the Rashomon Importance Distribution, aggregating importance scores from multiple models. Neither considered aggregating predictions before measuring importance. We prove (theoretically and empirically) that this overlooked ensemble alternative is more effective, identifying excess risk as the driving mechanism.
>
> Moreover, our analysis reveals that the conclusion is method-dependent: for CFI, the benefit of model-level ensembling is substantially weaker because the excess risk enters linearly rather than quadratically (section 4.3, confirmed in Figure 11). This nuance would not emerge without formal analysis.
>
> ## 3. UK Biobank: screening and biological interpretation
> **Screening**: While univariate screening could theoretically miss purely non-linear associations, our empirical results suggest that a 50-feature threshold retains the majority of the predictive signal in this context. Comparing predictive performance with 50 vs. 100 screened proteins yielded an $R^2$ of 0.64 and 0.65, respectively. This marginal gain suggests that the predictive information is mostly captured by the 50. The top-10 proteins also remain stable across both settings.
>
> **Biological interpretation**: The ensemble's top proteins align with known biological functions. The highest-ranked feature, Leptin, is a well-established hormone regulating body weight and obesity (Caro, 1996). Furthermore, only the ensemble confidently identifies IGFBP-2, a protective protein studied as a therapeutic target for obesity and diabetes (Haywood et al., 2019). We will add these references to the revised manuscript to contextualize the findings.
>
> **Biological ground truth**: To provide stronger evidence on a dataset with validated ground truth, we conducted a new experiment on the BRCA breast cancer dataset (see response to Reviewer ktp6). The ensemble strategy achieved significantly higher driver-gene ranking AUC than the sub-models strategy.
>
>
> ## 4. Notation fixes
> We will fix the sign discrepancy for $r_n$ in Appendix A.1(1) and use $[d]$ instead of $D$ in Theorem 4.10.
>
> ## References
> 1. Qu et al. TabICL: A Tabular Foundation Model for In-Context Learning on Large Data, ICML 2025
> 2. Fisher et al, All Models are Wrong, but Many are Useful, JMLR 2019
> 3. Donnelly et al, The Rashomon Importance Distribution: Getting RID of Unstable, Single Model-based Variable Importance, NeurIPS 2023
> 4. Caro et al, Leptin: the tale of an obesity gene Diabetes 1996
> 5. Haywood  et al, The insulin like growth factor and binding protein family: Novel therapeutic targets in obesity & diabetes 2019

---

> > ### Author Rebuttal · Reviewer_FnQq · 2026-04-02
> >
> > Thank you for the detailed response. The additional experiments and clarifications have addressed my main concerns regarding the experimental scope and novelty. I have adjusted my score.

---

### Decision · Program_Chairs · 2026-04-30

**Decision:**

Accept (regular)

**Comment:**

All reviewers agree that this paper takes a refreshing approach to a long-standing problem. During the rebuttal, the authors could address a few minor misunderstandings and, importantly, present additional results that convinced the reviewers of the practical value of the proposed approach. Overall, the reviewers recommend the paper for acceptance, and I completely agree. I do recommend the authors to include the clarifications and additional results along the lines of those presented and discussed during the rebuttal into the camera-ready copy. Of all changes, I would emphasize the significance of the contribution and clarifying that the paper addresses an actively studied problem but explores it in an opposite direction compared to what is common in the literature.